


# Thermal acclimation of leaf photosynthetic traits in an evergreen woodland, consistent with the co-ordination hypothesis

Henrique Fürstenau Togashi[1], Iain Colin Prentice[1,2], Owen K. Atkin[3,4], Craig Macfarlane[5], Suzanne M. Prober[5], Keith J. Bloomfield[3], Bradley John Evans[6]

[1]Department of Biological Sciences, Macquarie University, North Ryde, NSW 2109, Australia
[2]AXA Chair of Biosphere and Climate Impacts, Department of Life Sciences, Imperial College London, Silwood Park Campus, Buckhurst Road, Ascot SL5 7PY, UK
[3]Division of Plant Sciences, Research School of Biology, Australian National University, Canberra, Australia
[4]ARC Centre of Excellence in Plant Energy Biology, Research School of Biology, Australian National University, Canberra,
Australia
[5]CSIRO Land and Water, Private Bag 5, Wembley WA 6913, Australia
[6]Faculty of Agriculture and Environment, Department of Environmental Sciences, The University of Sydney, NSW 2006, Australia

*Correspondence to*: Henrique Fürstenau Togashi (henriquetogashi@gmail.com)

**Abstract.** Ecosystem models commonly assume that key photosynthetic traits, such as carboxylation-capacity measured at a standard temperature, are constant in time. The temperature responses of modelled photosynthetic/respiratory rates then depend entirely on enzyme kinetics. Optimality considerations suggest this assumption may be incorrect. The 'co-ordination hypothesis' (that Rubisco- and electron-transport limited rates of photosynthesis are co-limiting under typical daytime conditions) predicts instead that carboxylation ($V_{cmax}$) and light-harvesting ($J_{max}$) capacities, and mitochondrial respiration in
the dark ($R_{dark}$), should acclimate so that they increase with growth temperature – but less steeply than their instantaneous response rates. To explore this hypothesis, photosynthetic measurements were carried out on woody species during the warm and the cool seasons in the semi-arid Great Western Woodlands, Australia, under broadly similar light environments. A consistent linear relationship between $V_{cmax}$ and $J_{max}$ was found across species. $V_{cmax}$, $J_{max}$ and $R_{dark}$ increased with temperature, but values standardized to 25°C declined. The $c_i$:$c_a$ ratio increased slightly with temperature. The leaf $N$:$P$ ratio
was lower in the warm season. The slopes of the relationships of log-transformed $V_{cmax}$ and $J_{max}$ to temperature were close to values predicted by the co-ordination hypothesis, but shallower than those predicted by enzyme kinetics.

## 1 Introduction

Net photosynthetic $CO_2$ uptake ($A_{net}$) depends on temperature in all vegetation models, but models commonly disregard possible acclimation of the parameters determining $A_{net}$ to temporal variations in the growth environment. There are plentiful
data on the instantaneous (minutes to hours) temperature responses of photosynthetic uptake (Hikosaka et al., 2006; Sage and Kubien, 2007; Way and Sage, 2008), but data on the responses of photosynthetic traits on ecologically relevant time scales (days to years) are scarce (Lin et al., 2013). Nonetheless there is evidence that temperature responses of biochemical





processes are a function of plant growth temperature, and not just instantaneous temperature: from studies comparing species (Miyazawa and Kikuzawa, 2006; Kattge and Knorr, 2007) and experiments or serial measurements on single species (Medlyn et al., 2002b; Onoda et al., 2005). Neglecting acclimation to the growth temperature could lead to incorrect model estimates of the responses of primary production and ecosystem carbon budgets to climate change.

Farquhar et al. (1980) provided the standard model to predict photosynthetic responses to environment in $C_3$ plants. The model describes photosynthesis as instantaneously determined by the slower of two biochemical rates: the carboxylation of *RuBP* (ribulose-1,5-bisphosphate), dependent on Rubisco (ribulose-1,5-bisphosphate carboxylase/oxygenase) activity ($V_{cmax}$); and electron transport for *RuBP* regeneration in the Calvin cycle, which is dependent on light intensity and the capacity of the electron transport chain ($J_{max}$). Both rates are influenced by intercellular $CO_2$ concentration ($c_i$), which in turn

is partially regulated by stomatal conductance ($g_s$). Unlimited mesophyll conductance (Miyazawa and Kikuzawa, 2006; Lin et al., 2013) remains the standard implementation of the Farquhar model that should result in close or 'apparent' values of $V_{cmax}$ and $J_{max}$ as long as $K_c$ and $K_o$ (the Michaelis-Menten coefficients for carboxylation and oxygenation, respectively) are correct. These are highly variable both within and between species (Wullschleger, 1993), ranging through two to three orders of magnitude. Despite the accepted importance of these parameters for predicting rates of net $CO_2$ exchange in natural

ecosystems, a full understanding of how seasonal changes in the environment affect these parameters is lacking.

The 'coordination hypothesis' (Chen et al., 1993; Field and Mooney, 1986; Maire et al., 2012) predicts that values of $V_{cmax}$ and $J_{max}$ should acclimate, in time as well as in space, in such a way that carboxylation and *RuBP* regeneration are co-limiting under average daytime conditions. This has been suggested often in the literature, both as a prediction based on optimality considerations (e.g. Von Caemmerer and Farquhar, 1981; Haxeltine and Prentice, 1996; Dewar, 1996) and as an

empirical observation, consistent with the finding that carboxylation should be limiting at saturating light – as has been shown to apply in the great majority of cases (De Kauwe et al., 2015). The co-ordination hypothesis makes a number of testable predictions regarding photosynthetic properties of trees experiencing large seasonal variations in growth temperature, including an increase in $V_{cmax}$ with rising growth temperature – because at higher daily temperatures, greater Rubisco activity is required to match any given rate of photosynthesis. On the other hand, it predicts that values of $V_{cmax}$ and

$J_{max}$ normalized to 25°C – and also the leaf N per unit area (Maire et al., 2012) – should decline with increasing daily average temperature, as the quantities of proteins needed to maintain a given level of photosynthetic activity decline more steeply with temperature than the predicted increases in $V_{cmax}$ and $J_{max}$. These predictions were strongly supported by experiments on tree species grown at different temperatures (Scafaro *et al.* 2017), who also demonstrated that the acclimation of $V_{cmax}$ involves changes in Rubisco amount and the relative allocation of leaf N to Rubisco, rather than being a function of the

Rubisco activation state.

Much less is known about how acclimation of $R_{dark}$ (leaf respiratory $CO_2$ release in darkness) is linked to the coordination hypothesis, although $R_{dark}$ is also expected to acclimate to temperature. Acclimation results in homeostasis of $R_{dark}$ in plants grown at different temperatures, when measured at their respective growth temperatures (Larigauderie and Körner, 1995; Atkin and Tjoelker, 2003) and also results in $R_{dark}$ (at a standard temperature) increasing upon cold




acclimation and declining upon acclimation to warmer temperature (Reich et al., 2016).  Growth temperature dependent changes in $R_{dark}$ at a standard temperature can occur over periods of 1-3 days (Atkin et al., 2000; Bolstad et al., 2003; Lee et al., 2005; Zaragoza-Castells et al., 2007; Armstrong et al., 2008). An example of acclimation is a recent data synthesis of global patterns in leaf $R_{dark}$ (Atkin et al., 2015) which showed that geographic variation in $R_{dark}$ at growth temperature from

the tropics to the tundra is much smaller than would be expected on the basis of enzyme kinetics.  In that study, rates of rates of leaf $R_{dark}$ at a standard temperature increases with decreasing growth temperature across the globe (Atkin et al., 2015), with the global spatial patterns in $R_{dark}$ being consistent with acclimation of $R_{dark}$ to global patterns in growth temperature (Slot and Kitajima, 2015; Vanderwel et al., 2015).

Many models assume optimality criteria for stomatal behaviour, in which carbon assimilation is traded off against

water loss. Prentice et al. (2014) provided field evidence supporting the 'least-cost' hypothesis, which states that plants adopt an optimal $c_i$:$c_a$ ratio that minimizes the combined costs per unit carbon assimilation of maintaining the capacities for carboxylation ($V_{cmax}$) and water transport. This hypothesis predicts, *inter alia*, that the $c_i$:$c_a$ ratio should increase with the Michaelis-Menten coefficient for Rubisco-limited photosynthesis ($K$) and decrease with vapour pressure deficit ($D$).  The $c_i$:$c_a$ ratio is expected to increase with temperature, due to lower water viscosity (reducing water costs) and higher

photorespiration (increasing carboxylation costs) (Prentice et al., 2014), while declining with $D$ (Prentice et al., 2011).

In the current study, we present leaf-level measurements carried out during the warm and the cool seasons in the semi-arid environment of the Great Western Woodlands of southwestern Australia. By sampling during both seasons, we were able to compare observations under broadly overlapping light conditions but at contrasting temperatures. We are not aware of any previous study that has tested whether seasonal temperature acclimation is consistent with the co-ordination

hypothesis. We explore the idea by comparing the field-observed relationships of each trait to temperature with the theoretical acclimation of photosynthetic traits (as predicted by the coordination hypothesis) and with the alternative, i.e. the relationship of each trait to temperature to be expected if it were controlled only by enzyme kinetics (i.e. no acclimation).

## 2 Materials and Methods

### 2.1 Site

Eight representative woody species were studied at the Great Western Woodlands SuperSite (17°07'S, 145°37'E) approximately 70 km north-west of Kalgoorlie. The area has a semi-arid climate and the vegetation is a well preserved mosaic of temperate woodland, shrubland and mallee. The mean annual precipitation (*MAP*) for 1970-2013 was 380 mm (Hutchinson, 2014b). The average precipitation is slightly higher during summer months, but this rain often falls during short periods as intense storms. Mean annual temperature (*MAT*) is 20°C (Hutchinson, 2014c, a). Mean monthly daily temperature

minima range between 6°C and 18°C and maxima between 17°C and 35°C. The area is not prone to large shifts in temperature/vpd within days (e.g. cold fronts). Data for daily temperature and shortwave radiation were obtained from the flux tower (Australian and New Zealand Flux Research and Monitoring, www.ozflux.org.au). All trees were sampled within



a 5 km radius from the tower. The species studied were the evergreen angiosperm trees *Eucalyptus clelandii*, *E. salmonophloia*, *E. salubris* and *E. transcontinentalis* and the shrub *Eremophila scoparia;* the nitrogen-fixing leguminous tree *Acacia aneura* and shrub *A. hemiteles*; and one gymnosperm tree (*Callitris columellaris*).

**2.2 Gas exchange measurements**

We measured 109 *A-$c_i$* curves altogether during the warm season (late March/early April) and the cool season (late August/early September). The same individual plants were sampled in both seasons. A portable infrared gas analyser (*IRGA*) system (LI-6400; Li-Cor, Inc., Lincoln, NB, USA) was used. Sunlit terminal branches from the top one-third of the canopy were collected and immediately re-cut under water (Domingues et al., 2010). One of the youngest fully expanded leaves, attached to the cut branch, was placed in the leaf chamber. Measurements in the field were taken with the chamber block

temperature close to the air ambient temperature. The $CO_2$ partial pressure in the chamber for the *A-$c_i$* curves proceeded stepwise down from 400 to 35, back to 400 and then up to 2000 μmol mol$^{-1}$. Prior to the measurements, we tested plants to determine appropriate light-saturation levels. The photosynthetic photon flux density (PPFD) adopted for measurement ranged between 1500 and 1800 μmol m$^{-2}$ s$^{-1}$. After measuring the *A-$c_i$* curves over about 35 minutes, light was set to zero for five minutes before measuring $R_{dark}$. Following Domingues et al. (2010), we discarded 23 A-$c_i$ curves in which $g_s$ declined to

very low levels (resulting in 86 curves being used in further analyses), adversely affecting the calculation of $V_{cmax}$. TPU (triose phosphate utilization) limitation (Sharkey et al., 2007) was not considered, as it would be unlikely to occur at our field temperatures of above 17 ˚C. The data are available via the TERN Supersites Data Portal (Togashi et al., 2015). Reported ratios of $c_i$:$c_a$ relate to chamber conditions, with ambient $CO_2 \approx 400$ μmol mol$^{-1}$.

**2.3 Photosynthetic parameters and their temperature responses**

Apparent values of $V_{cmax}$ and $J_{max}$ were fitted using the Farquhar et al. (1980) model. Values were standardized to 25˚C ($V_{cmax25}$ and $J_{max25}$) using the *in vivo* temperature dependencies given in Bernacchi et al. (2001) and Bernacchi et al. (2003). Following Bernacchi et al. (2009), we used the Arrhenius equation to describe the temperature responses of $V_{cmax}$, and $J_{max}$:

$$parameter = parameter_{25} \exp\left[\frac{(T_k-298.15)\Delta H_a}{R' T_k 298.15}\right] \tag{1}$$

where $\Delta H_a$ is the activation energy (J mol$^{-1}$), $R'$ is the universal gas constant (8.314 J mol$^{-1}$ K$^{-1}$) and $T_k$ is leaf temperature

(K).

To derive $R_{dark25}$ we applied a temperature-dependent $Q_{10}$ (fractional change in respiration with a 10˚C increase in temperature) equation in which $Q_{10}$ declines with increasing leaf temperature (Atkin and Tjoelker, 2003):

$$R_{dark25} = R_{dark}\left(3.09 - 0.043\left[\frac{T_2+T_1}{2}\right]\right)^{\left[\frac{T_2-T_1}{10}\right]} \tag{2}$$

where 3.09 and 0.04 are empirical values, $T_2$ (25˚C) and $T_1$ are leaf temperatures (˚C) for $R_{dark25}$ and $R_{dark}$ respectively.




### 2.4 Nutrient analyses

After completion of each $A$-$c_i$ curve, leaves were retained to determine leaf area, dry mass, and mass-based nitrogen ($N$) and phosphorus ($P$) concentrations (mg g$^{-1}$). Leaves were sealed in plastic bags containing moist tissue paper to prevent wilting. Leaf area was determined using a 600 dots/inch flatbed top-illuminated optical scanner and *Image J* software
(http://imagej.nih.gov/ij/). Leaves were dried in a portable desiccator for 48 hours, to be preserved until the end of the campaign. Subsequently, in the laboratory, leaves were oven-dried for 24 hours at 70°C and the dry weight determined (Mettler-Toledo Ltd, Port Melbourne, Victoria, Australia). Leaf mass per unit area ($LMA$; g m$^{-2}$) was calculated from leaf area and dry mass. $N_{mass}$ and $P_{mass}$ were obtained by Kjeldahl acid digestion of the same leaves (Allen et al., 1974). The leaf material was digested using sulphuric acid 98% and hydrogen peroxide 30%. Digested material was analyzed for N and P
using a flow injection analyser system (LaChat QuikChem 8500 Series 2, Lachat Instruments, Milwaukee, WI, USA). Area-based $N$ and $P$ values (mg m$^{-2}$) were calculated as products of $LMA$ and $N_{mass}$ or $P_{mass}$.

### 2.5 Statistical analyses

All statistics were performed in R (R Core Team, 2012). For graphing we used the *ggplot2* package (Wickham, 2010). $V_{cmax}$, $V_{cmax25}$, $J_{max}$, $J_{max25}$, $R_{dark}$, $R_{dark25}$, $N_{area}$ and $P_{area}$ data were log$_{10}$-transformed, unless otherwise indicated, to approximate a
normal distribution of values. The ratio $c_i$:$c_a$ was not transformed because of its small variance and approximately normal distribution in this study. Linear regression ($lm$) was used to test dependencies among parameters. Slopes and elevations of regressions were compared using standardized major axis regression with the *smatr* package (Warton et al., 2006). The Welch two-sample $t$-test was used to test pairwise differences in traits (e.g. differences between the warm season and the cool season measurements). Generalized linear models ($glm$) were used to test acclimation to temperature across species,
using species and temperature as predictors.

### 2.6 Predicted responses to temperature

Log-transformed $V_{cmax}$, $J_{max}$ and $R_{dark}$ were regressed against leaf temperature measured in the field. All regressions were fitted across species, with species identity included as a predictor ($glm$) to control for species effects on parameter values at a given temperature. We compared the slopes of these regressions with theoretically derived values based on alternative
hypotheses: (1) based on enzyme kinetics (without acclimation), and (2) based on the co-ordination hypothesis for $V_{cmax}$, $J_{max}$ and $R_{dark}$ and the least-cost hypothesis for $c_i$:$c_a$. The derivations are given in Appendix A. Although enzyme activation energy ($\Delta H_a$) can vary among species and with temperature (Von Caemmerer, 2000), for simplicity we adopted the same $\Delta H_a$ for all species and temperatures. This approximation could affect interspecies comparisons of $\Delta H_a$-dependent parameters but should not substantially interfere with the comparison of theoretical and fitted slopes.



## 3 Results

### 3.1 Relationships among photosynthetic parameters

When measured at near-ambient air temperature, species-average $V_{cmax}$ values ranged across seasons from 44.4 to 105 μmol m$^{-2}$ s$^{-1}$, $J_{max}$ from 77.4 to 160 μmol m$^{-2}$ s$^{-1}$, $R_{dark}$ from 1.16 to 3.14 μmol m$^{-2}$ s$^{-1}$ and $c_i{:}c_a$ from 0.39 to 0.60 (at ambient $CO_2 \approx$

400 μmol mol$^{-1}$). At the prevailing air temperatures, $V_{cmax}$, $J_{max}$ and $R_{dark}$ were systematically higher in the warm season, while their values standardised to 25˚C were higher in the cool season (Fig. 1). The $c_i{:}c_a$ ratio also exhibited significantly higher average values in the warm season in six out of eight species (not shown).

$V_{cmax}$ and $J_{max}$ were closely correlated across species within and across seasons (Fig. 2). The warm and the cool seasons regression equations relating $V_{cmax}$ and $J_{max}$ were statistically indistinguishable. The warm and the cool seasons

slopes of regressions forced through the origin are shown in Fig. 2. Regressions between $V_{cmax}$ and $J_{max}$ for the warm and the cool seasons together yielded $J_{max} = 0.84 \ V_{cmax} + 55.2$ ($p < 0.05$, slope standard error = 0.2). There were positive correlations between $R_{dark25}$ and both $V_{cmax}$ and $J_{max}$ across seasons, as well as strong negative correlations between species-average $c_i{:}c_a$ ratios and both $V_{cmax}$ and $J_{max}$ across seasons (Fig. 3). Log transformed $R_{dark}$ was only correlated to $V_{cmax}$ and $J_{max}$ ($p < 0.05$) for the individual species *E. salmonophloia* and *C. columellaris*.

### 3.2 Leaf gas exchange trait responses to temperature

Based on data from all species together, $V_{cmax}$, $J_{max}$ and $R_{dark}$ all increased with leaf temperature, while the corresponding normalized (to 25˚C) values declined with leaf temperature ($p < 0.05$, Fig. 4). The $c_i{:}c_a$ ratio also increased slightly but significantly with leaf temperature, ($p < 0.05$; Fig. 5a). The ratio $J_{max}{:}V_{cmax}$ did not correlate with temperature based on the data from all species together, but it was negatively correlated with temperature for *E. salmonophloia*, *E. scoparia* and *C.*

*columellaris* (not shown). Excluding the two N-fixing species, and/or the one gymnosperm, from the dataset had no effect on these results.

The relationship between ambient air temperature and leaf temperature (˚C) was $T_{leaf} = 1.01 \ T_{air} + 0.35$ ($p < 0.05$, $R^2 = 0.96$). Regression slopes between photosynthetic parameters and $T_{air}$ showed no significant differences from those calculated using $T_{leaf}$, but the goodness of fit was weaker with $T_{air}$ than with $T_{leaf}$. We also fitted regressions using $T_{day}$ (the

daily mean temperature). Again the slopes did not change, but the goodness of fit was further reduced. The factor 'season' (included as a predictor in a *glm*, in addition to $T_{leaf}$) did not improve model fit.

Within individual species, we also found positive responses of $V_{cmax}$ and $J_{max}$ to temperature, and negative responses when the parameters were standardised to 25˚C (Fig. 6). The response of the $c_i{:}c_a$ ratio to leaf temperature was similar in most species (Fig. 5b). Within-species responses of $R_{dark}$ to leaf temperature were weaker and less consistent (Fig. 6),

suggesting that respiration had acclimated to a greater extent than was the case for $V_{cmax}$ and $J_{max}$.



Incoming shortwave radiation at the surface is used here as a proxy for PPFD. Daily values ranged from 90 to 256 W m$^{-2}$ and averaged 193 W m$^{-2}$. Averages for the warm and the cool seasons sampling periods were not significantly different. None of the photosynthetic parameters showed any relationship with shortwave radiation.

### 3.3 Leaf *N* and *P* relationships to photosynthetic traits and temperature

Area-based rates of leaf gas exchange traits were not systematically related to total leaf $N_{area}$ or $P_{area}$ (data not shown).  There was a positive relationship between *N* and *P* (by mass) taking all species together, and within three of the species ($p < 0.05$; Fig. 7). High values of the foliar *N:P* ratios (> 16) in seven out of eight species (Fig. 8) may suggest that *P* in this ecosystem is more limiting to growth than *N* (Westoby and Wright, 2006). *N:P* ratios declined with increasing temperature ($p < 0.05$). *A. aneura* and *A. hemiteles* presented the highest N:P as expected for N-fixing species.

### 3.4 Quantitative temperature responses

For $V_{cmax}$ and $J_{max}$, the fitted slopes with leaf temperature were shallower than the predicted 'kinetic' slopes by a margin that greatly exceeded their 95% confidence limits (Table 1). The 'kinetic' values are what would be expected if the activities of the relevant enzyme complexes remained constant with changing growth temperature. The co-ordination hypothesis predicts
much shallower 'acclimated' rate-temperature slopes. The acclimated slope for $V_{cmax}$ can be predicted from the temperature dependencies of *K* and the photorespiratory compensation point ($\Gamma^*$), as shown by Eq. (A3) in Appendix A. The acclimated slope for $V_{cmax}$ falls just marginally above the 95% confidence interval for the fitted slope. The acclimated slope of $J_{max}$ (see Appendix A for calculation) falls well within the 95% confidence interval for the fitted slope. For $R_{dark}$ the acclimated and kinetic slopes are closer together and both fall within the 95% confidence interval of the fitted slope.

There is no 'kinetic' response of $c_i{:}c_a$, but the least-cost hypothesis predicts a positive response to temperature. This was observed, although the fitted slope of the response to temperature was shallower than predicted. The observed response of $c_i{:}c_a$ to temperature is taken into account as a small correction to the acclimated slope of $V_{cmax}$ against temperature (Appendix A).

## 4 Discussion

### 4.1 Quantitative ranges of photosynthesis traits

Values of $V_{cmax}$, $J_{max}$, $R_{dark}$ and $c_i{:}c_a$ for the eight species measured here were within ranges commonly reported. $V_{cmax}$ and $J_{max}$ were generally lower than expected for desert species, but higher than typical values for mesic perennial species (Wullschleger, 1993). The values were also high compared with trees from savannas with twice the annual precipitation (Domingues et al., 2010). The $c_i{:}c_a$ ratios fall within the range typical for dry environments (Prentice et al., 2014).



### 4.2 Comparison between seasons

Our results were consistent with acclimation of photosynthetic traits to temperature as predicted by the coordination hypothesis. When measured at the prevailing ambient temperature, $V_{cmax}$, $J_{max}$ and $R_{dark}$ were all generally higher in the warm season than in the cool season, whereas values standardized to 25˚C were generally lower in the warm season than in the

cool season (Fig. 1). This is *prima facie* evidence for active seasonal acclimation, as the co-ordination hypothesis predicts lower allocation of $N$ to Rubisco and other photosynthetic enzymes at higher temperatures, offsetting the increase in enzyme activity with elevated temperature. Moreover, absence of acclimation should result in no relationship between $V_{cmax25}$ and leaf temperature, while the negative relationship found here is evidence for acclimation (Figs. 4 and 6).

Levels of leaf $N$ and $P$ have been reported to change seasonally (Medlyn et al., 2002a, and figure 8). We found a

reduction in the $N:P$ ratio in the warm season, consistent with a reduced allocation of $N$ to photosynthetic functions (Way and Sage, 2008). A reduction in total leaf $N$ does not necessarily indicate changes in $N$ allocation; however, a strong coupling between $N$ and photosynthesis has been widely observed, even though Rubisco accounts for only 10-30% of the total leaf $N$ (Evans, 1989). Furthermore, a reduction of leaf N in the warm season is unlikely to be caused by general growth dilution during an actively growing part of the year because: a) this is an environment with a year-round growing season; b)

PPFD during the periods of the field campaigns was similar; and c) the measured $V_{cmax}$ values were shown to be consistent with the coordination hypothesis, implying similar assimilation rates in the two seasons. We did not find significant relationships of photosynthetic traits to foliar $N$ or $P$ despite the study area being extremely limited in supplies of both nutrients (Prober et al., 2012).

The comparison of fitted and theoretically predicted slopes (Table 1) reveals not only that the responses of $V_{cmax}$,

$J_{max}$ and $R_{dark}$ to ambient temperature were smaller than would be predicted from enzyme kinetics alone, but also that the observed responses were close to, or (in the case of $J_{max}$) statistically indistinguishable from the responses predicted by the co-ordination hypothesis. The response of $c_i:c_a$ is in the same direction (positive) as the response predicted by the least-cost hypothesis, but only about half as large, probably due to the opposing effect (reduction in $c_i:c_a$) of greater vapour pressure deficits in the warm season than in the cool season (Prentice et al., 2014).

$V_{cmax}$ and $J_{max}$ were strongly and positively correlated across species (Figure 2), and the relationship did not shift significantly between seasons. Some studies have reported a lower $J_{max}:V_{cmax}$ ratio in warmer seasons compared to cooler seasons (Medlyn et al., 2002a; Lin et al., 2013). Our data show a tendency in this direction, but not enough to be significant (Fig. 2). $V_{cmax}$ and $J_{max}$ have previously been reported to increase seasonally with leaf temperature. In one recent study on six *Eucalyptus* species, measurements were taken at six temperature levels in winter, spring and summer; there was an increase

in $V_{cmax}$ and $J_{max}$ with air temperature in seasons with overlapping temperatures, and $V_{cmax25}$ was significantly higher in the winter than in the summer (Lin et al., 2013). Miyazawa and Kikuzawa (2006) obtained similar results in five evergreen broadleaved species. Our measurements yielded similar results (Figs. 4 and 6).




### 4.3 Links between photosynthetic activity, $R_{dark}$ and $c_i$:$c_a$

$V_{cmax}$, $J_{max}$ and $R_{dark}$ were positively correlated with leaf temperature across a wide range (cool season 17 to 27˚C; warm season 26 to 37˚C), both for the dataset as a whole and within individual species (Figs. 4 and 6). Photosynthetic capacity and respiratory flux are linked – in part - via the ATP (adenosine triphosphate) demand of sucrose synthesis and transport,

leading to the interdependence of chloroplast and mitochondrial metabolism (Krömer, 1995; Ghashghaie et al., 2003). The parallel temperature acclimation of $R_{dark}$ and $V_{cmax}$ illustrates this close relationship.

     Across all species, there was a strong negative relationship between $c_i$:$c_a$ and photosynthetic capacity. This is to be expected from the co-ordination hypothesis as the lower the $c_i$:$c_a$ ratio, the greater photosynthetic capacity ($V_{cmax}$ and $J_{max}$) required to achieve a given assimilation rate. $c_i$:$c_a$ ratios increased with temperature as predicted by the least-cost hypothesis,

but the slope of 0.006 (Fig. 5) based on all data is shallower than the predicted slope of 0.0131. This difference may reflect higher vapour pressure deficits in the warm season, which would be expected to close stomata and therefore act in the opposite way to the effect of temperature alone (Prentice et al., 2014).

### 4.4 Implications for modelling

One dynamic global vegetation model, the Lund-Potsdam-Jena (LPJ) model (Sitch et al., 2003) together with later models based on LPJ, formally assumes the co-ordination hypothesis (as well as the coupling between $R_{dark}$ and $V_{cmax}$) and thus implicitly allows photosynthetic parameters and leaf respiration to acclimate, on a daily to monthly time scale, to the seasonal course of climate. The same assumption is implicit in the simple primary production models developed by Wang et al. (2013; 2014) but data available to test this model assumption is scarce. Our study has presented field evidence supporting

predictions of the co-ordination hypothesis regarding the seasonal acclimation of $V_{cmax}$, $J_{max}$ and the allocation of $N$ in leaves. Terrestrial models that do not allow such acclimation may incorrectly represent the seasonal time course of carbon exchange at the plant and ecosystem levels.

### 5 Appendix A: Derivation of theoretical responses of photosynthetic traits to temperature

#### 5.1 Kinetic responses (no acclimation)

Temperature responses of many biological reaction rates are accurately described (within normal physiological ranges) by the Arrhenius equation, which can be written as:

$$\ln param\,(T)\ -\ \ln param\,(T_{ref}) \ =\ (\Delta H_a/R)\,(1/T_{ref} - 1/T) \tag{A1}$$

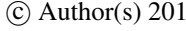


where $T$ is the measurement temperature and $T_{ref}$ is a reference temperature (K), $\Delta H_a$ is the activation energy of the reaction (J mol$^{-1}$ K$^{-1}$) and $R$ is the universal gas constant (8.314 J mol$^{-1}$ K$^{-1}$). This expression can be approximated by a simple exponential function as follows, by linearizing around $T_{ref}$:

$$\ln param\,(T) \;-\; \ln param\,(T_{ref}) \;\approx\; (\Delta H_a/R)\,(1/T_{ref}^2)\,\Delta T \tag{A2}$$

where $\Delta T = T - T_{ref}$. Thus, the slope of $\ln param\,(T)$ *versus* $T$ can be predicted from Eq. (A2), when $\Delta H_a$ is known. We set $T_{ref} = 298$ K (which is both conventional, and close to the median measurement temperature in our data set). We used the $\Delta H_a$ values based on *in vivo* measurements at 25˚C by Bernacchi et al. (2001) for $V_{cmax}$ (65 330 J mol$^{-1}$ K$^{-1}$). For $J_{max}$, we used the $\Delta H_a$ value based on *in vivo* measurements by Bernacchi et al. (2003) on plants that had been grown at 25˚C (43 900 J mol$^{-1}$ K$^{-1}$). For $R_{dark}$, we used $\Delta H_a$ (50230 J mol$^{-1}$ K$^{-1}$) calculated from Eq. (2).

**5.2 Acclimated responses**

The acclimated response of $V_{cmax}$ according to the co-ordination hypothesis is obtained by setting the Rubisco- and electron-transport limited rates of photosynthesis to be equal. To simplify matters, we disregarded the curvature of the response of electron transport to PPFD, giving:

$$V_{cmax} \;=\; \varphi_0\,I_{abs}\,(c_i + K)/(c_i + 2\Gamma*) \tag{A3}$$

where $\varphi_0$ is the intrinsic quantum efficiency of photosynthesis, $I_{abs}$ is the absorbed PPFD, $K$ is the effective Michaelis-Menten coefficient for carbon fixation and $\Gamma*$ is the photorespiratory compensation point. The sensitivity of $V_{cmax}$ to temperature can then be calculated from the derivative of (A3):

$$\partial \ln V_{cmax}/\partial T = (1/V_{cmax})\,\partial V_{cmax}/\partial T \;=\; (\partial c_i/\partial T)\,[1/(c_i + K) - 1/(c_i + 2\Gamma*)] +$$
$$+ (\partial K/\partial T)\,[1/(c_i + K)] - 2\,(\partial \Gamma*/\partial T)[1/(c_i + 2\Gamma*)] \tag{A4}$$

20        We evaluated Eq. (A4) at 25˚C and $c_i = 200$ µmol mol$^{-1}$ (approximately the median of our observed values of $c_i$), using the temperature dependencies of $K$ and $\Gamma*$ from Bernacchi et al. (2001). The temperature dependency of $K$ was determined from those of the constituent terms $K_c$ and $K_o$ (the Michaelis-Menten coefficients for carboxylation and oxygenation, respectively) as given by Bernacchi et al. (2001), using similar methods.

        Again for simplicity, we assumed an approximation of the theoretical acclimated slope for $J_{max}$ that is simply the
acclimated slope of $V_{cmax}$, minus the difference between the kinetic slopes of $V_{cmax}$ and $J_{max}$. We assumed that $R_{dark}$ (on acclimation) should simply be equal to a constant fraction of $V_{cmax}$, implying the same acclimated temperature response for $R_{dark}$ as for $V_{cmax}$.



The least-cost hypothesis (Prentice et al., 2014) provides an optimal value for $c_i{:}c_a$, denoted by $\chi_o$, such that:

$$\chi^* = \chi_o/(1 - \chi_o) = \sqrt{(bK/1.6aD)} \tag{A5}$$

where $b$ is the (assumed constant) ratio of $R_{dark}$ to $V_{cmax}$ and $a$ is the cost of maintaining the transpiration pathway. Both $K$ and $a$ are temperature-dependent; $a$ because it is proportional to the viscosity of water, $\eta$. Holding vapour pressure deficit ($D$) constant, we can obtain an expression for $\partial\chi^*/\partial T$:

$$\partial\chi^*/\partial T = (\chi^*/2)\,(\partial \ln K/\partial T - \partial a/\partial T) \tag{A6}$$

and from Eqs. (A5) and (A6), with the help of the chain rule,

$$\partial\chi_o/\partial T = (\chi_o/2)\,(1 - \chi_o)\,(\partial \ln K/\partial T - \partial a/\partial T) \tag{A7}$$

We evaluated Eq. (A7) at $T = 25°C$, $\chi_o = 0.5$ and $c_a = 400$ ppm using known temperature dependencies of $K$ and $\eta$.

**Data sets**

Data can be requested at the Terrestrial Ecosystem Research Network (TERN) data discovery portal http://portal.tern.org.au (Prober *et al.* 2015).

**Authors contribution**

H.F.T, I.C.P, O.K.A, C.M, and S.M.P planned and designed the research. H.F.T performed all research and data analysis. H.F.T measured all the field data used in the experiment. H.F.T wrote the first draft; all authors commented on subsequent versions and assisted with data interpretation.

**Competing interests**

The authors declare that they have no conflict of interest.

**Acknowledgements**

This research was funded by the Terrestrial Ecosystem Research Network (TERN), Macquarie University and the Australian National University. H.F. Togashi is supported by an international Macquarie University International Research Scholarship (iMQRES). Prentice, Evans, and Togashi have been funded by the Ecosystem Modelling and Scaling Infrastructure (eMAST, http://www.emast.org.au, part of TERN). TERN and eMAST are supported by the Australian Government through



the National Collaborative Research Infrastructure Strategy (NCRIS). Owen Atkin acknowledges the support of the Australian Research Council (DP130101252 and CE140100008). The Australian SuperSites Network and OzFlux (part of TERN), the CSIRO Land and Water Flagship, and the Western Australia Department of Environment and Conservation support the Great Western Woodlands Supersite. *N* and *P* were analysed in the Department of Forestry, ANU. We are

grateful to Jack Egerton (ANU), Li Guangqi (Macquarie), Lingling Zhu (ANU), Danielle Creek (University of Western Sydney), Lasantha Weerasinghe (ANU), Lucy Hayes (ANU) and Stephanie McCaffery (ANU) for help with fieldwork and/or N and P digestions. We thank Santi Sabaté (Universitat Autònoma de Barcelona) and Maurizio Mencuccini (University of Edinburgh) for comments that helped to improve this paper. This paper is a contribution to the AXA Chair Programme in Biosphere and Climate Impacts and the Imperial College initiative on Grand Challenges in Ecosystems and

the Environment.

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



**Tables**

Table 1: Linear regression slopes ($K^{-1}$) and their 95% confidence intervals for log-transformed photosynthetic traits, with species included as a factor. The values are compared to 'kinetic' slopes (as expected in the absence of acclimation) and 'acclimated' slopes, as predicted by the co-ordination hypothesis for $V_{cmax}$, $J_{max}$ and $R_{dark}$ and the least-cost hypothesis for $c_i{:}c_a$. See Appendix A for the calculation of kinetic and acclimated slopes.

|  | $V_{cmax}$ | $J_{max}$ | $R_{dark}$ | $c_i{:}c_a$ |
|---|---|---|---|---|
| *Fitted* | 0.0328 | 0.0251 | 0.0514 | 0.0060 |
|  | (± 0.0158) | (±0.0108) | (±0.0164) | (±0.0033) |
| *Kinetic* | 0.0885 | 0.0628 | 0.0675 | n/a |
| *Acclimated* | 0.0493 | 0.0236 | 0.0494 | 0.0131 |

**Figures**





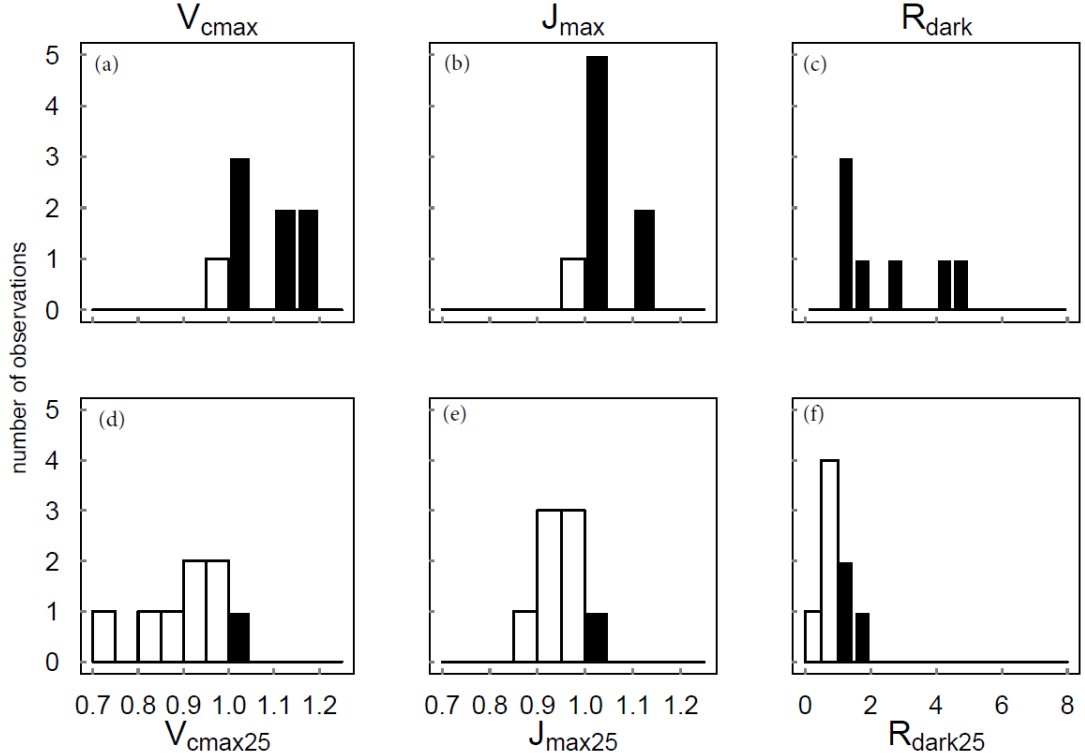

**Figure 1: Species distribution warm:cool seasons ratios of log$_{10}$-transformed $V_{cmax}$, $J_{max}$ and $R_{dark}$ measured at ambient temperature (top panels) and standardized to 25°C (bottom panels). Ratios > 1 are shown as black bars, ratios < 1 as white bars (n = 8).**




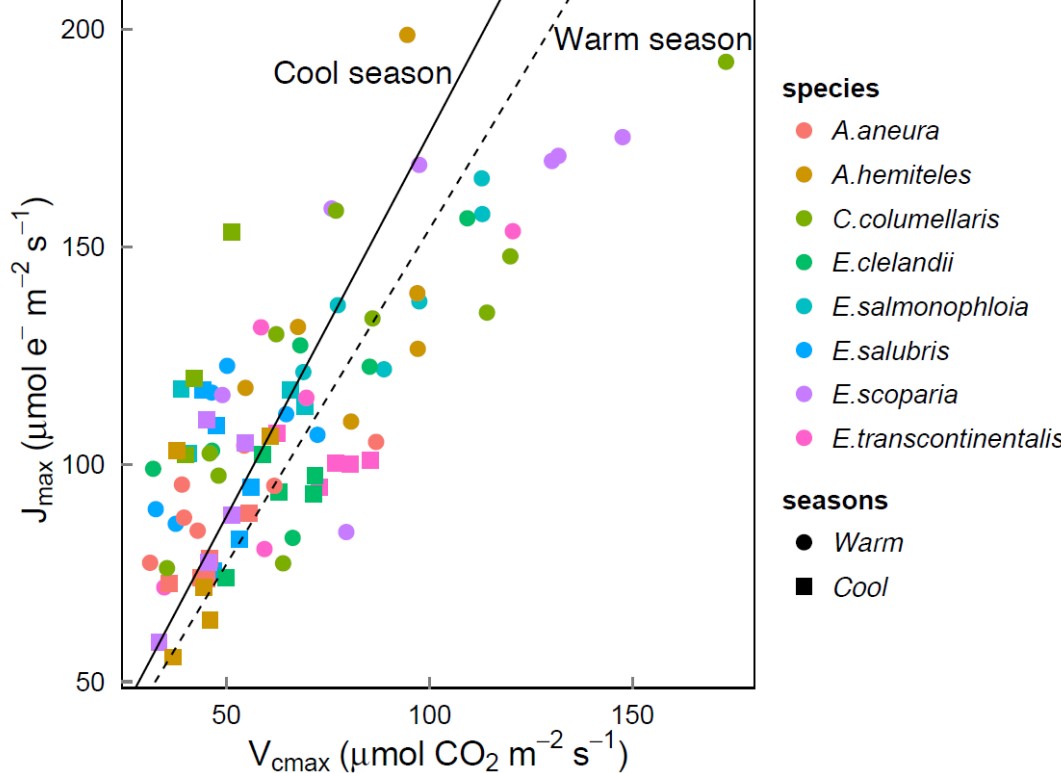

**Figure 2: Linear regressions forced through the origin between $J_{max}$ (μmol m$^{-2}$s$^{-1}$) and $V_{cmax}$ (μmol m$^{-2}$s$^{-1}$) for individual trees of eight species in the warm season (circles, dashed line, slope = 1.58) and the cool season (squares, solid line, slope = 1.79). Both regressions are significant ($p < 0.05$). Each point represents one $A$-$c_i$ curve (n=86).**




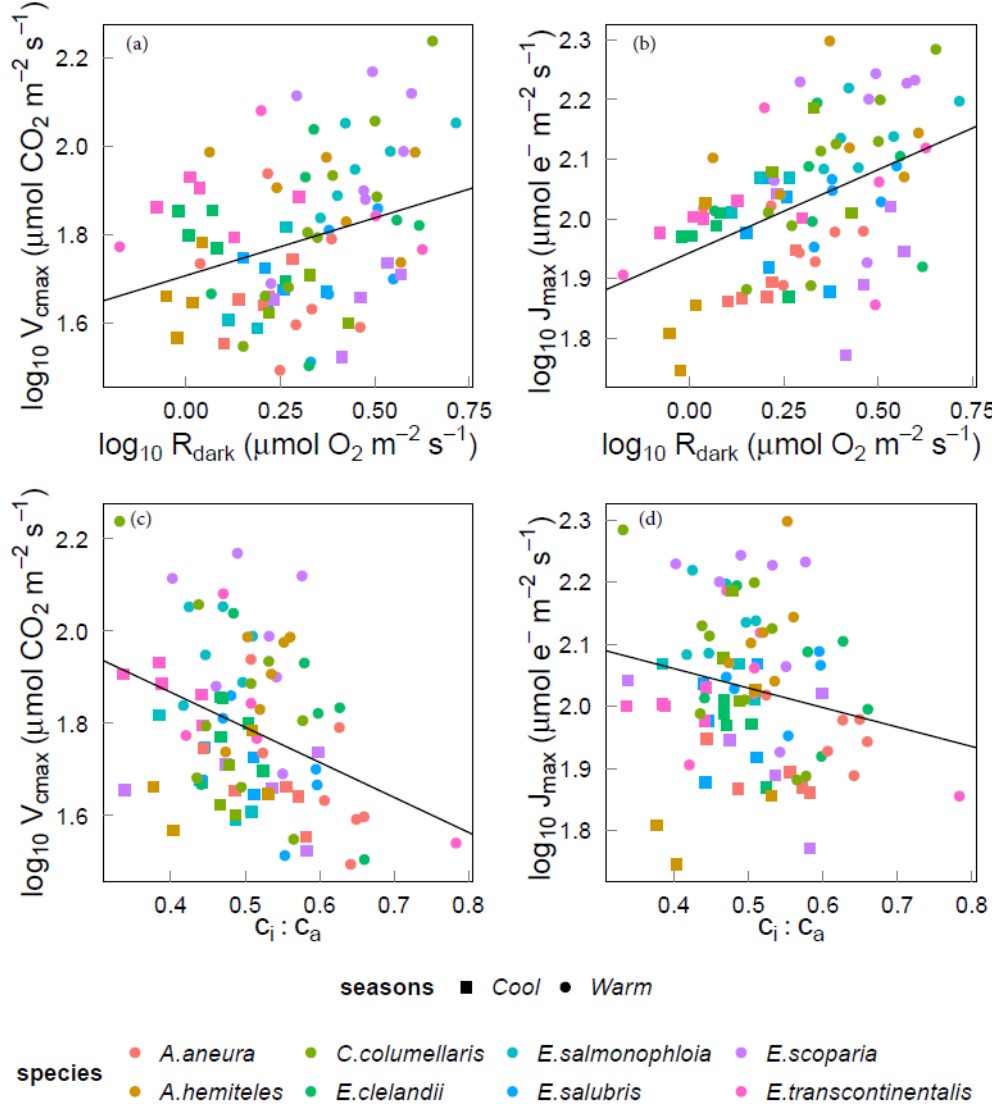

Figure 3: Linear regressions of individual trees by season (n=16) and species (n=8) for photosynthetic capacity, $V_{cmax}$ and $J_{max}$ with $R_{dark}$ (µmol m$^{-2}$s$^{-1}$) and the $c_i$:$c_a$ ratio at ambient $CO_2 \approx 400$ µmol mol$^{-1}$ ($p < 0.05$). $V_{cmax}$, $J_{max}$ and $R_{dark}$ were log$_{10}$ transformed; $c_i$:$c_a$ was not. Each point represents one $A$-$c_i$ curve (n=86).





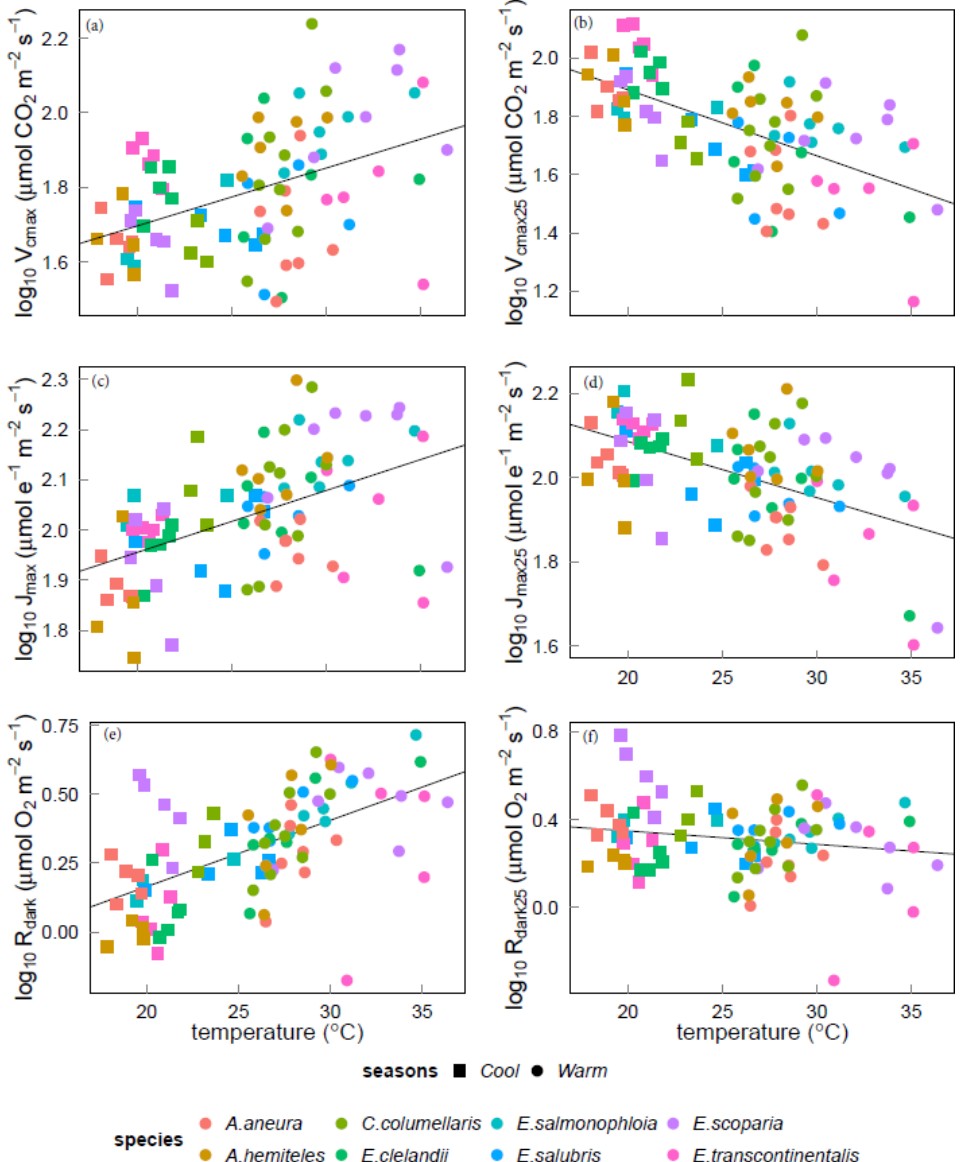

**Figure 4: Bivariate linear regressions ($p < 0.05$) of $\log_{10}$ transformed $V_{cmax}$, $V_{cmax25}$, $J_{max}$, $J_{max25}$, $R_{dark}$ and $R_{dark25}$ ($\mu$mol m$^{-2}$s$^{-1}$) *versus* leaf temperature ($T_{leaf}$, °C). Each point represents one $A$-$c_i$ curve (n=86).**





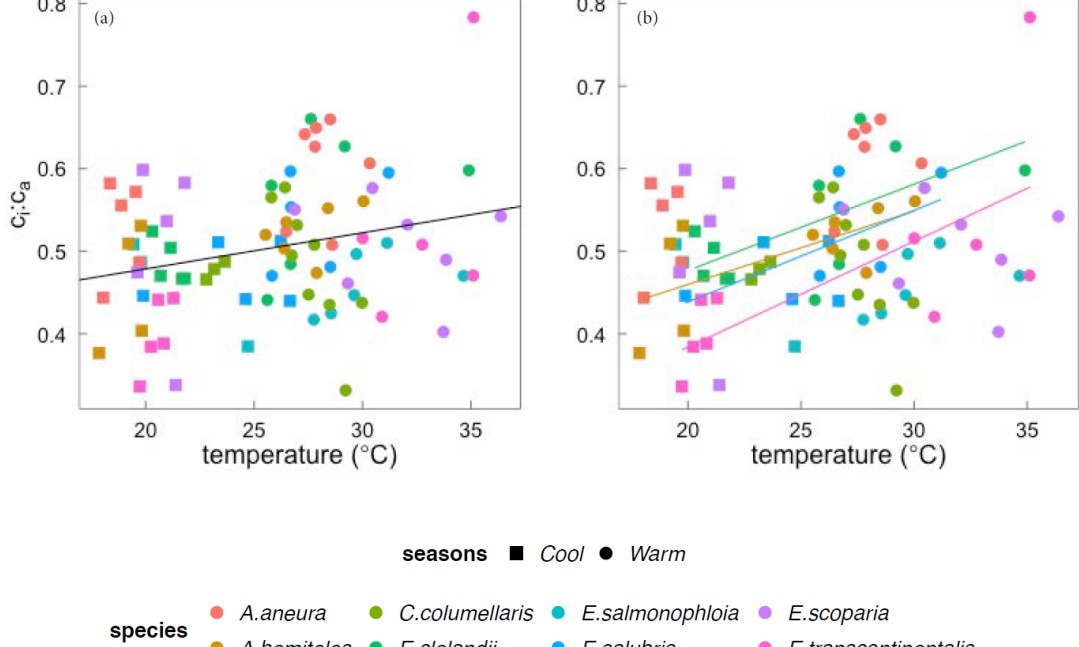

Figure 5: Bivariate linear regressions of the $c_i$:$c_a$ ratio (at ambient $CO_2 \approx 400$ µmol mol$^{-1}$) versus temperature ($T_{leaf}$, °C) for individual trees considering all data (a) and within species (b). Only significant regressions ($p < 0.05$) are shown. Each point represents one $A$-$c_i$ curve (n=86).





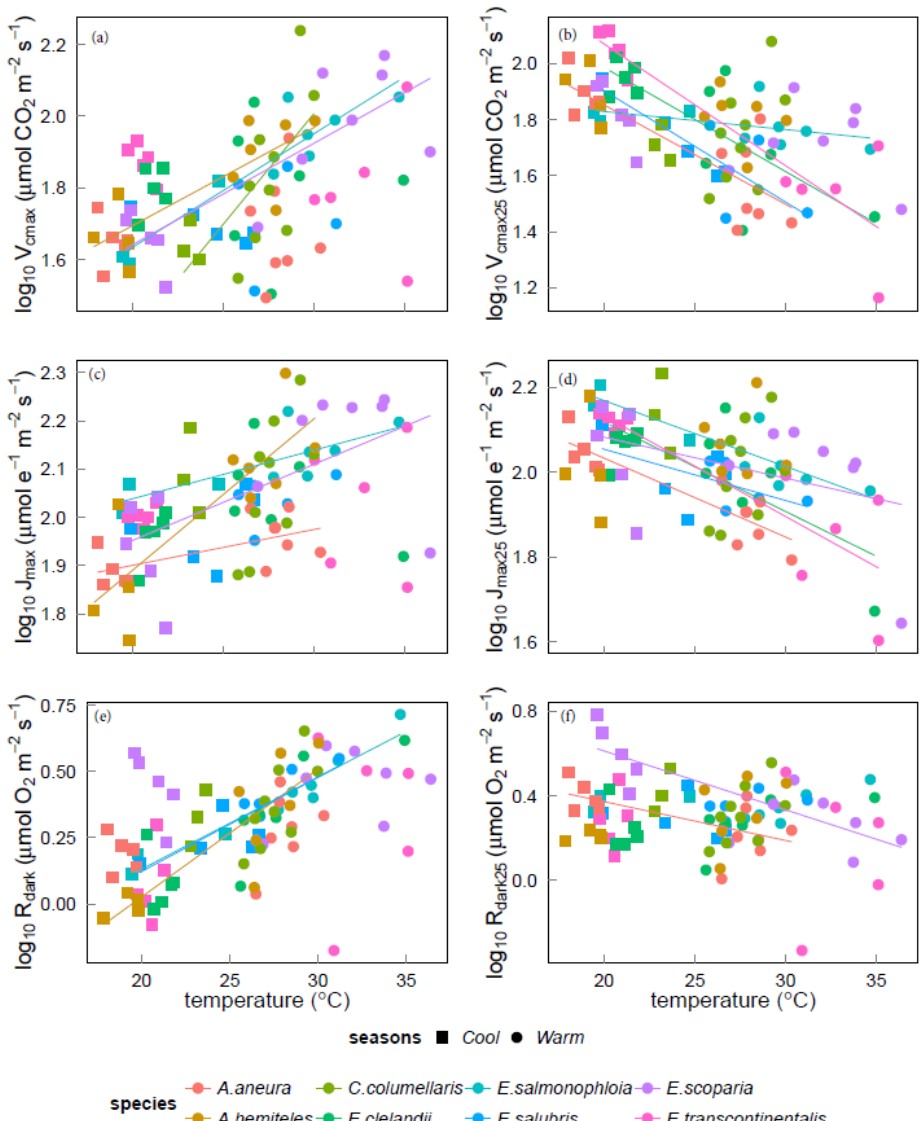

**Figure 6: Bivariate linear regressions of log$_{10}$ transformed $V_{cmax}$, $V_{cmax25}$, $J_{max}$, $J_{max25}$, $R_{dark}$ and $R_{dark25}$ (µmol m$^{-2}$ s$^{-1}$) *versus* leaf temperature ($T_{leaf}$, °C) within species ($p < 0.05$). Only significant regressions ($p < 0.05$) are shown. Each point represents one $A$-$c_i$ curve (n=86).**



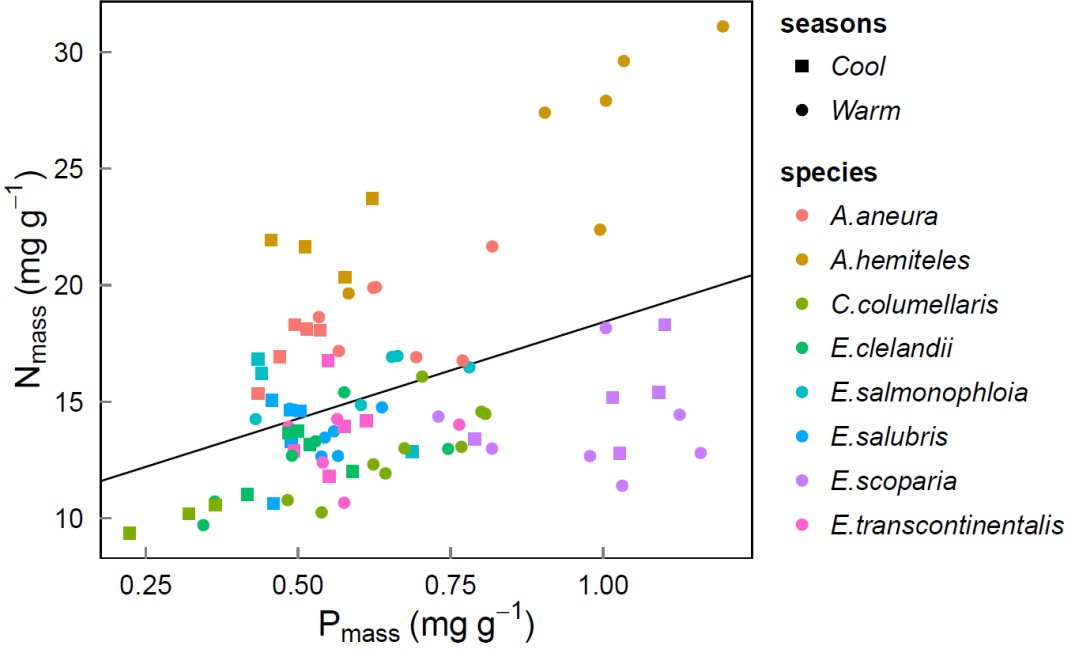

**Figure 7:** Bivariate linear regression of $N_{mass}$ (mg g$^{-1}$) *versus* $P_{mass}$ (mg g$^{-1}$) for all data (black line, slope = 0.33, R$^2$ = 0.17). Each point represents one *leaf* (n=86).




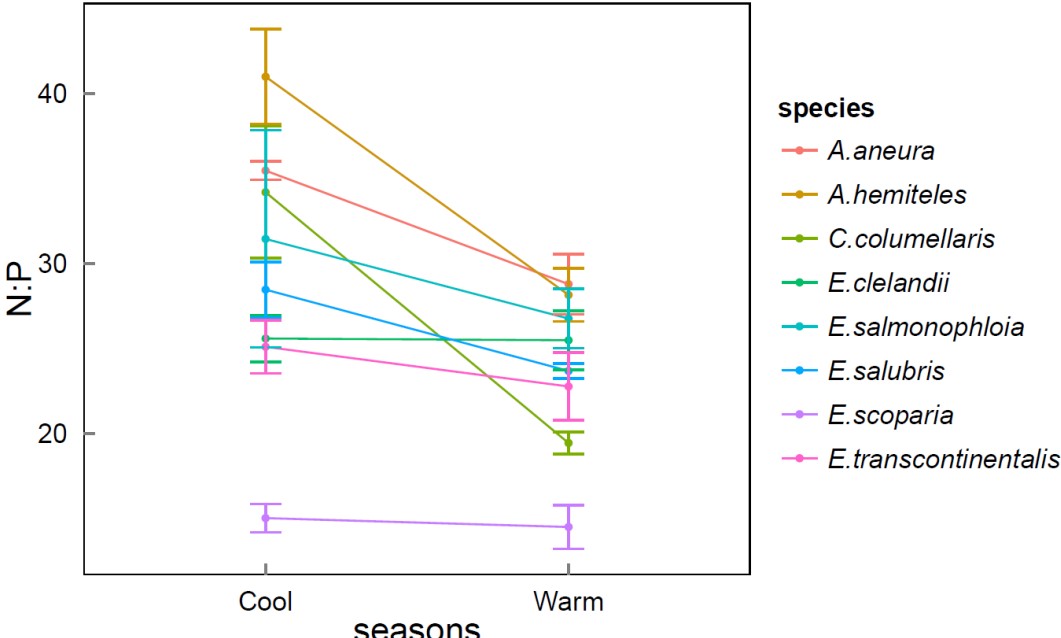

**Figure 8: Changes in the average foliar *N:P* ratio for each species between the cool and the warm seasons. Standard errors shown.**