# Peer review of "Thermal acclimation of leaf photosynthetic traits in an evergreen woodland, consistent with the coordination hypothesis"

_Biogeosciences, 2017_

## Referee Comment (RC1) · Anonymous Referee #1 · 4 Dec 2017

**General comments**

This paper looks at the seasonal acclimation of photosynthetic parameters and whether this acclimation is consistent with the coordination hypothesis. To this end, the authors make photosynthetic measurements in two distinct seasons for several species in an Australian evergreen woodland and then compare the measured temperature response with the theoretical expectation. The findings are particularly important because photosynthetic parameters are often considered invariant in time in earth system models and also because seasonal photosynthetic measurements the same location are rare.

I find that while the results from the photosynthetic measurements are interesting and

well analysed and discussed, the theoretical analysis and the link between measurements and theory needs more discussion, in particular in relation to the many linear assumptions made, all of which are hidden in the appendix.

**Specific comments**

*Data analysis and regressions*

The conclusions relating to acclimation and coordination are based on the slopes of regression lines of photosynthetic variables to temperature but there is insufficient detail in the paper relating to the results of the statistical analysis. Figures 3-5 are presented without any goodness of fit metrics or p-values for the individual lines. In addition, the authors assume a linear relationship between the log10 values of each variable and temperature, an assumption which is detailed in appendix A but not sufficiently discussed in the main text.

*Coordination hypothesis*

The coordination hypothesis states that the Rubisco and electron transport limited rates are co-limiting under average conditions, which is generally taken to mean that there is a change in the $J_{max25}$ to $V_{cmax25}$ ratio and implicitly a change in nitrogen allocation inside the leaf. The authors make a linear approximation to solve for this co-limitation (eq. A3). This approximation removes the parameter $J_{max25}$ from the calculation and its slope with temperature is calculated assuming proportionality to the slope of $V_{cmax25}$ and a ratio of the biochemical temperature response. While these approximations can be justified, I believe that a further discussion is needed as the resulting equations are difficult to match with the coordination hypothesis as this is generally understood. I would also suggest including all the equations in the main body of the text since they are necessary to the central message of the paper

The authors report the slope of the log10 of each measured parameter with temperature and compare this to the theoretical equivalent slope (Table 1) to reach the conclusion that the coordination hypothesis is valid. The more usual approach would be to calculate the theoretically predicted values of the photosynthetic parameters and plot these together with the measured values. The authors' approach is scientifically valid but given the multiple approximations and log values I find it hard to follow.

Also, the fitted slopes for all parameters are calculated as log10(parameter) vs. temperature, while the theoretical slopes are ln(parameter) vs. temperature. I would suggest that the authors check their calculations and verify that these slopes are equivalent.

*Leaf nitrogen variation*

Changes in $V_{cmax}$ values alone do not verify the coordination hypothesis - these can be caused either by acclimation or by changes in total leaf nitrogen. According to Fig. 8 there are large differences in the leaf N for some species, which can be caused by a number of factors apart from temperature acclimation, especially leaf ageing. I would be interested to see how the ratio of $V_{cmax}$ (and/or $J_{max}$) to leaf N changes seasonally, which would give a better indication of photosynthetic coordination.

*Leaf respiration*

While acclimation of respiration is a well documented and important process it is unclear how this links to the coordination hypothesis. Here the authors hypothesised that dark respiration scales linearly with Vcmax and will therefore follow the coordination hypothesis as well, but this is not necessarily the case in either models or reality and a better justification of why the variation in dark respiration should be linked with photosynthetic co-limitation is needed.

**Technical comments**

The authors should decide whether we are talking about 'coordination' or 'co-ordination'.

---

## Short Comment (SC1) · 18 Dec 2017

Togashi et al. collected data for evergreen tree species from southwestern Australia and found that temperature acclimation of these species are consistent with the coordination hypothesis. The central motivation of this work is due to the fact that numerous ecosystem models assume that photosynthetic traits are constant in time and perhaps coordination hypothesis, which is based on optimality, would be useful for the models.

While this study is interesting, I would like the authors to clarify the following:

a) Based on their data or via model simulations, suggest how the ecosystem models

can be improved. That is, if you were to use an ecosystem model, how would the parameters that you measured change with time in the model. In my view, coordination hypothesis has already been implemented in some ecosystem models.

b) You have the seasonal data and you just connect two points in Fig. 8. First in my view, this does not seem right. It would be nice to show better the temporal variation of the parameters for these evergreen species. My main concern here is to specify how much is the variation in the parameters of these evergreen species due to the different seasons e.g. 10%, 20%, etc.

Best

---

## Referee Comment (RC2) · Anonymous Referee #2 · 3 Jan 2018

General comments

The authors have performed a suite of gas exchange measurements and leaf biochemical analyses in a suite of species across two seasons. The novel angle to the paper is to present this data in the light of the coordination hypothesis. The work is well done and interesting.

Specific comments

My main comment is that the discussion is very thin. It could use more substance and less reiterating the results. What do you make of the considerable spread in the data? Why do many species in Figure 6 not show the expected response, even if the pooled

data does? There's a lot more here to discuss than is currently covered.

There are a number of studies that have measured Vcmax and Jmax at multiple times across a season in the literature (Baldocchi has a few, for example). These should be acknowledged in the intro. Similarly, there should be a citation to Way and Yamori 2013 who found no change in Vcmax25 in a meta-analysis of plants grown at different temperatures.

Why was Rdark measured after only 5 mins in the dark? This is usually measured after at least 20 and often 30 minutes of darkness to get a true estimate of dark respiration.

What VPD were the measurements made at? If the summer VPD is higher, gs will be reduced, which will lower the Ci/Ca ratio, but this isn't necessarily a temperature effect per se.

Figure 2 - why were the fits forced through the origin and how does this affect the slopes? Is it a minimal effect?

Lastly, while I appreciate the use of the log-transformed data to get linear slopes, I'd like to see the "real" data, at least in the SI. This makes it much easier to see the values measured and compare the data with the majority of other studies that report Vcmax and Jmax values against leaf temperature.

Technical comments

Page 2, Line 13 - please clarify what "these" refers to - Vcmax and Jmax, yes?

Page 9, Line 7 - the relationship between Ci/Ca and photosynthetic capacity could also be because higher photosynthetic capacity (at a constant gs) reduces Ci. Cause and effect can't be determined.

If all the gas exchange is determined with a Licor IRGA, how are the parameters being reported in units of electrons and O2? Jmax and Rdark should be in units of CO2 per area per time.

---

## Referee Comment (RC3) · Anonymous Referee #3 · 9 Jan 2018

Review of 'Thermal acclimation of leaf photosynthetic traits in an evergreen woodland, consistent with the co-ordination hypothesis 'by Fuersteau Togashi and co-authors

This study analyses gas exchange data taking at two seasons in a semi-arid evergreen woodland in Australia to test the presence of thermal acclimation of photosynthetic capacity, leaf dark respiration and Ci:Ca using optimality principles.

The paper is overall well written and clear. I have few comments & questions that follow in order of relevance.

-What is the role of phenology / specifically leaf age here, there is a need to discuss this either in the introduction and or discussion, i.e. there might be confounding phenological and thermal acclimation effects in the presented results. What is the life time of a leaf in this semi-arid evergreen woodland?

-Related to the above, the manuscript provides an explanation of changes in N:P ratios from cold to warm season in Fig 8 however it does not explain how these changes happened, how did leaf N and P changed and how this might be related to leaf age? It would be good to add a plot showing individual values of leaf N and Leaf P in the cold and warm season.

-Equation 1 is used to estimate Vcmax and Jmax at 25C. During the warm period (unclear time of day A-Ci curves where taken) Vmax at T could be either in the optimum or beyond the optimum temperatures, thus it is possible that the peaked temperature response might be more appropriate. If this was the case, how is this likely to affect the results? Also, how does the choice of Ha (Medlyn et al 2002) value affects the results. According to Hikoska et al (2006) there is a relationship between activation energy of Vcmax and growth temperature.

-Similar comments apply to the use of equation A2 to determine the slope of Vcmax and temperature presented in Table 1 under the kinetic approach. Is the slope sensitive to the choice of Ha but most important are the slope values robust when estimated with the peaked temperature response for Vcmax and Jmax.

-Leaf dark respiration measurements were taken after only 5 minutes of leaves being in the dark. Protocol for Rd estimates is at least 30 min in the dark (Atkin et al 2000; Atkin et al 1998) as it takes about 15-20 minutes for post-illumination respiration to stabilize with time increasing with decreasing temperature. How does this affect your measurements of Rdark and acclimation results?

-On the implications for modelling section it would be very relevant to apply the Kattge & Knorr (2007) formulations and compare to your data set and predictions by the optimization approach used in this study. Is the data from this study consistent with the Vcmax25 prediction derived by Scafaro et al (2017)

-Either in the introduction or in the methodology, it would be good to include a graphic explaining the change in temperature responses to illustrate what acclimation is, i.e. temperature response shifts forward and therefore values at 25 C decline, you could illustrate also where in the curve are the leaf temperature values are during the cold and warm season.

-It would be useful to include a figure of the mean diurnal cycle of air temperature during the warm and cold seasons but also provide an idea of when the A-CI curves were taken and under which RH, VPD conditions. If RH & VPD conditions differ, what are the implications

Minor comments P10, L 25 Can you clarify in the text why the acclimated slope of Jmax to leaf temperature was estimated as the acclimated slope of Vcmax minus the difference of the kinetic slopes of Vcmax and Jmax (this might also be affected by peaked temperature response)

P3 L2 –can elaborate here and explain homeostasis

P6 L22 Is this Tleaf measured by the Licor or an independent measurement? If yes would be good to mention it in the methods section

P7 L26-29 These values were not really shown as it was all logged transformed, would be nice to show the data. The sentences comparing values to dessert plants and mesic perennial species could be more specific and include typical values for those vegetation types otherwise is all very generic and less informative.

P8 L6 but 'lower allocation of N to Rubisco' has not been demonstrated here

P8 L9, need to mention the role of leaf age /phenology, maybe here good to show N values change and use this to support some of the sentences on this paragraph

References Atkin et al (2000) Plant Physiology, 122, 915–924.

Atkin et al (1998) Australian Journal of Plant Physiology, 25, 437–443

Medlyn et al (2002), Plant, Cell and Environment, 25, 1167–1179.

Hikosaka et al (2006), Journal of Experimental Botany, 57, 291–302.

Kattge & Knorr (2007) Plant, Cell & Environment, 30, 1176–1190.

Scafaro et al (2017) Global Change Biology 23, 2783–2800.
* * *

---

## Author Comment (AC1) · 2 Mar 2018

We thank the reviewers for comments and suggestions that have helped to improve our manuscript. Our response is organised by addressing each comment one by one.

Anonymous Referee #1

AR1.1. I find that while the results from the photosynthetic measurements are interesting and C1 well analysed and discussed, the theoretical analysis and the link between measurements and theory needs more discussion, in particular in relation to the many linear assumptions made, all of which are hidden in the appendix.

- We have moved the equations to the main text and provided further discussion there. This discussion now includes the rationale for the linear assumptions.

AR1.2. The conclusions relating to acclimation and coordination are based on the slopes of regression lines of photosynthetic variables to temperature but there is insufficient detail in the paper relating to the results of the statistical analysis. Figures 3-5 are presented without any goodness of fit metrics or p-values for the individual lines.

- We believe this comment refers to Figures 5 and 6. Only significant individual lines (p < 0.05) are shown in Figures 5 and 6, as is described in the Figure captions and in the Results section (3.1 and 3.2).

AR1.3. In addition, the authors assume a linear relationship between the log10 values of each variable and temperature, an assumption which is detailed in appendix A but not sufficiently discussed in the main text.

- Linear regressions are used, because the theoretical equations relating log-transformed traits to temperature are linear. This should now be clear from the revised text, in which the equations are presented up-front.

AR1.4. The coordination hypothesis states that the Rubisco and electron transport limited rates are co-limiting under average conditions, which is generally taken to mean that there is a change in the $Jmax25$ to $Vcmax25$ ratio and implicitly a change in nitrogen allocation inside the leaf. The authors make a linear approximation to solve for this co-limitation (eq. A3). This approximation removes the parameter $Jmax25$ from the calculation and its slope with temperature is calculated assuming proportionality to the slope of $Vcmax25$ and a ratio of the biochemical temperature response. While these approximations can be justified, I believe that a further discussion is needed as the resulting equations are difficult to match with the coordination hypothesis as this is generally understood.

- The reviewer makes an insightful point here. The key evidence is the change in

Vcmax. It is not completely clear what the co-ordination hypothesis predicts about Jmax. The reviewer may be right that "it is generally taken" that the co-ordination hypothesis is all about the partitioning of leaf N to Jmax versus Vcmax. But the co-ordination hypothesis merely states that the two limiting rates of photosynthesis tend to be equal under average conditions. Limitation by Jmax is not normally reached under natural conditions; at low light photosynthesis is limited by J (not Jmax) and at high light it becomes limited by Vcmax. So the first-order prediction of the co-ordination hypothesis is that Vcmax should acclimate to the average light conditions. A subsidiary hypothesis is then required to predict the ratio of Jmax to Vcmax (Wang et al., 2017). We hope we have now made this clear in the text, while avoiding too much distracting complexity.

AR1.5. I would also suggest including all the equations in the main body of the text since they are necessary to the central message of the paper

- Done.

AR1.6. The authors report the slope of the log10 of each measured parameter with temperature and compare this to the theoretical equivalent slope (Table 1) to reach the conclusion that the coordination hypothesis is valid. The more usual approach would be to calculate the theoretically predicted values of the photosynthetic parameters and plot these together with the measured values. The authors' approach is scientifically valid but given the multiple approximations and log values I find it hard to follow.

- We agree that plots are easier to follow than tables; however, we do present partial residual plots – please bear in mind that to maximize statistical power we have analysed the data together in a generalised linear model, using species as factors (as explained in the Methods – Statistical Analysis section, and the captions). We tried various ways to present the data and we found that the approach we have adopted here was the most accessible.

AR1.7. Also, the fitted slopes for all parameters are calculated as log10(parameter)

vs. temperature, while the theoretical slopes are ln(parameter) vs. temperature. I would suggest that the authors check their calculations and verify that these slopes are equivalent.

- The data presented has now being standardise to ln.

AR1.8. Changes in Vcmax values alone do not verify the coordination hypothesis - these can be caused either by acclimation or by changes in total leaf nitrogen. According to Fig. 8 there are large differences in the leaf N for some species, which can be caused by a number of factors apart from temperature acclimation, especially leaf ageing. I would be interested to see how the ratio of Vcmax (and/or Jmax) to leaf N changes seasonally, which would give a better indication of photosynthetic coordination.

- We do not entirely agree with this suggestion. The co-ordination hypothesis predicts that Vcmax should acclimate so as to just use the available light under average conditions (as pointed out above) and that implies (as pointed out by Maire et al., 2012) that there is an optimal leaf N for any given set of average environmental conditions. It is not just a hypothesis about the allocation of a given amount of N to different functions within the leaf. Nonetheless, in response to this comment, we have carried out additional analyses on how the ratio of Vcmax (and Vcmax25) to Narea varies with growth temperature. The results are generally very similar to the analysis of Vcmax (and Vcmax25), but they are significant for more species (7/8: all except the gymnosperm Callitris columelaris, for both Vcmax and Vcmax25). We have clarified the point, and referred to this additional analysis, in our revised text. See also the graphs attached.

AR1.9. While acclimation of respiration is a well documented and important process it is unclear how this links to the coordination hypothesis. Here the authors hypothesised that dark respiration scales linearly with Vcmax and will therefore follow the coordination hypothesis as well, but this is not necessarily the case in either models or reality and a better justification of why the variation in dark respiration should be linked with
photosynthetic co-limitation is needed.

- The reviewer is correct to indicate that (as with Jmax) to predict the acclimation of Rdark to temperature from the co-ordination hypothesis requires an additional hypothesis. We have tried out the simplest, i.e. that Rdark remains proportional to Vcmax. This logic has been clarified.

AR1.10. The authors should decide whether we are talking about 'coordination' or 'co-ordination'.

- The spelling 'co-ordination' has been replaced by 'coordination'.

—- Anonymous Referee #2

AR2.1. My main comment is that the discussion is very thin. It could use more substance and less reiterating the results. What do you make of the considerable spread in the data? Why do many species in Figure 6 not show the expected response, even if the pooled data does? There's a lot more here to discuss than is currently covered.

- We have expanded the discussion to cover these points.

AR2.2. There are a number of studies that have measured Vcmax and Jmax at multiple times across a season in the literature (Baldocchi has a few, for example). These should be acknowledged in the intro. Similarly, there should be a citation to Way and Yamori 2013 who found no change in Vcmax25 in a meta-analysis of plants grown at different temperatures.

- We have included the suggested citations, and commented on the issue raised.

AR2.3. Why was Rdark measured after only 5 mins in the dark? This is usually measured after at least 20 and often 30 minutes of darkness to get a true estimate of dark respiration.

- This was a time-saving compromise to allow four or five replicate curves per machine per day, based on our experience that stable results are commonly obtained after 5

minutes. Moreover, this quick estimate should still be superior to the common practice of deriving Rdark as one of the parameters in a curve-fitting routine. We have added a brief comment on this.

AR2.4. What VPD were the measurements made at? If the summer VPD is higher, gs will be reduced, which will lower the Ci/Ca ratio, but this isn't necessarily a temperature effect per se.

- There is indeed a systematic difference between VPD in the two seasons. The average VPD value during the warm season was 1.13 kPa, and during the cool season 0.45 kPa. The average leaf-to-air VPD (i.e. corrected to leaf temperature) during the warm season was 2.5 kPa, and during the cool season 1.44 kPa. However, there was very little difference in stomatal conductance at light saturation: (0.064 – 0.082 – 0.101) (lower quartile – median – upper quartile) mol m–2 s–1 for the warm season, and (0.057 – 0.078 – 0.085) mol m–2 s–1 for the cool season. We have added a note on this point.

AR2.5. Figure 2 - why were the fits forced through the origin and how does this affect the slopes? Is it a minimal effect?

- Both slopes are significant if not forced through origin. However, an intercept for this relationship is (a) extrapolated and (b) makes no biological sense. We have added a note on this.

AR2.6. Lastly, while I appreciate the use of the log-transformed data to get linear slopes, I'd like to see the "real" data, at least in the SI. This makes it much easier to see the values measured and compare the data with the majority of other studies that report Vcmax and Jmax values against leaf temperature.

- The real values are presented in the Results section and in Figure 2. Raw data are also available on the TERN portal, as mentioned in the text.

Technical comments

AR2.7. Page 2, Line 13 - please clarify what "these" refers to - Vcmax and Jmax, yes?

- Yes. We have clarified this now.

AR2.8. Page 9, Line 7 - the relationship between Ci/Ca and photosynthetic capacity could also be because higher photosynthetic capacity (at a constant gs) reduces Ci. Cause and effect can't be determined.

- We have added a note on this alternative explanation.

AR2.9. If all the gas exchange is determined with a Licor IRGA, how are the parameters being reported in units of electrons and O2? Jmax and Rdark should be in units of CO2 per area per time.

- We have amended the units as suggested.

—- Anonymous Referee #3

AR3.1. What is the role of phenology / specifically leaf age here, there is a need to discuss this either in the introduction and or discussion, i.e. there might be confounding phenological and thermal acclimation effects in the presented results. What is the life time of a leaf in this semi-arid evergreen woodland? -Related to the above, the manuscript provides an explanation of changes in N:P ratios from cold to warm season in Fig 8 however it does not explain how these changes happened, how did leaf N and P changed and how this might be related to leaf age? It would be good to add a plot showing individual values of leaf N and Leaf P in the cold and warm season.

- Three factors militate against a role for phenology and leaf ageing in our results. First, the generally long (5-10 years) lifetime of leaves in this ecosystem, meaning that ageing proceeds slowly; second, there is continuing turnover of leaves through the year in this ecosystem; and second, we sampled the youngest fully expanded leaves in both seasons. We have added a comment in the text to this effect.

AR3.2. Equation 1 is used to estimate Vcmax and Jmax at 25C. During the warm

period (unclear time of day A-Ci curves where taken) Vmax at T could be either in the optimum or beyond the optimum temperatures, thus it is possible that the peaked temperature response might be more appropriate. If this was the case, how is this likely to affect the results? Also, how does the choice of Ha (Medlyn et al 2002) value affects the results. According to Hikoska et al (2006) there is a relationship between activation energy of Vcmax and growth temperature. Similar comments apply to the use of equation A2 to determine the slope of Vcmax and temperature presented in Table 1 under the kinetic approach. Is the slope sensitive to the choice of Ha but most important are the slope values robust when estimated with the peaked temperature response for Vcmax and Jmax.

- In theory Topt can affect the calculation of Vcmax and Jmax. However, Medlyn et al. (2002), Kattge & Knorr (2007), and others have found very good correlations between Topt and mean daily temperature. We measured A-ci curves between morning and early afternoon, avoiding the hottest part of the day. Therefore, it is unlikely that any of our measurements were carried out above the optimum temperature. We have added a comment to this effect. Regarding the activation energy of Vcmax, there have been reports of a sensitivity to growth temperature but any such effect appears to be small and some studies have failed to find any such effect (e.g. Kattge & Knorr, 2007). No information was available on Ha for the species sampled and so (in common with much of the ecophysiological literature) we adopted a generic in vivo value. We recognize that the activation energy of Jmax is much more variable with growth conditions.

AR3.3. Leaf dark respiration measurements were taken after only 5 minutes of leaves being in the dark. Protocol for Rd estimates is at least 30 min in the dark (Atkin et al 2000; Atkin et al 1998) as it takes about 15-20 minutes for post-illumination respiration to stabilize with time increasing with decreasing temperature. How does this affect your measurements of Rdark and acclimation results?

- This was a time-saving compromise (see our response under AR2.3 above).

AR3.4. On the implications for modelling section it would be very relevant to apply the Kattge & Knorr (2007) formulations and compare to your data set and predictions by the optimization approach used in this study.

- See our response to AR3.2 above.

AR3.5. Is the data from this study consistent with the Vcmax25 prediction derived by Scafaro et al (2017)

- Yes. This was mentioned in the original text, but the reference was missing in the bibliography. We have rectified this omission.

AR3.6. Either in the introduction or in the methodology, it would be good to include a graphic explaining the change in temperature responses to illustrate what acclimation is, i.e. temperature response shifts forward and therefore values at 25 C decline, you could illustrate also where in the curve are the leaf temperature values are during the cold and warm season.

- We have provided the suggested graphic.

AR3.7. It would be useful to include a figure of the mean diurnal cycle of air temperature during the warm and cold seasons but also provide an idea of when the A-CI curves were taken and under which RH, VPD conditions. If RH & VPD conditions differ, what are the implications.

- We have now provided information in the text on the diurnal cycle of temperature in both seasons, and the timing of our measurements. Regarding the potential effects of VPD, please see our response under AR2.4 above.

AR3.8. P10, L 25 Can you clarify in the text why the acclimated slope of Jmax to leaf temperature was estimated as the acclimated slope of Vcmax minus the difference of the kinetic slopes of Vcmax and Jmax (this might also be affected by peaked temperature response)

- We have now explained what we are doing here in the revised text. It is certainly a simplification – see our response to AR1.4 above.

AR3.9. P3 L2 –can elaborate here and explain homeostasis

- We have changed the wording here to be more explicit.

AR3.10. P6 L22 Is this Tleaf measured by the Licor or an independent measurement? If yes would be good to mention it in the methods section

- Tleaf is the leaf temperature as measured by the LiCor, and Tair is the air temperature. We have noted this in the methods section.

AR3.11. P7 L26-29 These values were not really shown as it was all logged transformed, would be nice to show the data.

- The values are presented in the Results section and in Figure 2. Raw data are also available on the TERN portal, as mentioned in the text. Since the manuscript already includes eight Figures and one Table, we prefer to keep the reader's attention focused on the main results.

AR3.12. The sentences comparing values to dessert plants and mesic perennial species could be more specific and include typical values for those vegetation types otherwise is all very generic and less informative.

- We have include some example values.

AR3.13. P8 L6 but 'lower allocation of N to Rubisco' has not been demonstrated here.

- We agree, but this is presented in the text here as a prediction, not as a fact! We have added some words to clarify this further.

AR3.14. P8 L9, need to mention the role of leaf age /phenology, maybe here good to show N values change and use this to support some of the sentences on this paragraph.

- Please see our response to AR 3.1.

—- Short Comment #1

SC1.1 Based on their data or via model simulations, suggest how the ecosystem models can be improved. That is, if you were to use an ecosystem model, how would the parameters that you measured change with time in the model. In my view, coordination hypothesis has already been implemented in some ecosystem models.

- We have added some wording to address this point in the Discussion.

SC1.2. You have the seasonal data and you just connect two points in Fig. 8. First in my view, this does not seem right. It would be nice to show better the temporal variation of the parameters for these evergreen species. My main concern here is to specify how much is the variation in the parameters of these evergreen species due to the different seasons e.g. 10%, 20%, etc.

- We were not monitoring the species through a whole year, and so it is not possible to provide what is asked for here. However, we have provided some indication of the relative magnitude of seasonal changes in parameters.
* * *
Fig. 1.

[Figure]

**species**
- *A.aneura*
- *A.hemiteles*
- *C.columellaris*
- *E.clelandii*
- *E.salmonophloia*
- *E.salubris*
- *E.scoparia*
- *E.transcontinentalis*

**Fig. 2.**

---

## Author Response (AR2)

[revised manuscript text omitted]

R Core Team: R:A language and environment for statistical computing. R Foundation for Statistical Computing, Vienna, Austria. http://www.R-project.org/, 2012.

Quebbeman, J. A. and Ramirez, J. A.: Optimal allocation of leaf-level nitrogen: Implications for covariation of $V_{cmax}$ and $J_{cmax}$ and photosynthetic downregulation, Journal of Geophysical Reseasch Biogeosciences, 121, 2464–2475, 2016.

Reich, P. B., Sendall, K. M., Stefanski, A., Wei, X., Rich, R. L., and Montgomery, R. A.: Boreal and temperate trees show strong acclimation of respiration to warming, Nature, 531, 633–636, 2016.

Sage, R. F., and Kubien, D. S.: The temperature response of $C_3$ and $C_4$ photosynthesis, Plant, Cell & Environment, 30, 1086-1106, 2007.

Scafaro, A. P., Negrini, A. C. A., O'Leary, B., Rashid, F. A. A., Hayes, L., Fan, Y., Zhang, Y., Chochois, V., Badger, M. R., and Millar, A. H.: The combination of gas-phase fluorophore technology and automation to enable high-throughput analysis of plant respiration, Plant Methods, 13, 16, 2017.

Sharkey, T. D., Bernacchi, C. J., Farquhar, G. D., and Singsaas, E. L.: Fitting photosynthetic carbon dioxide response curves for $C_3$ leaves, Plant, cell & environment, 30, 1035-1040, 2007.

[revised manuscript text omitted]

seasons ■ Cool ● Warm

species ● *A.aneura*  ● *C.columellaris*  ● *E.salmonophloia*  ● *E.scoparia*
● *A.hemiteles*  ● *E.clelandii*  ● *E.salubris*  ● *E.transcontinentalis*

**Figure 5: Bivariate linear regressions ($p < 0.05$) of natural log transformed $V_{cmax}$, $V_{cmax25}$, $J_{max}$, $J_{max25}$, $R_{dark}$ and $R_{dark25}$ ($\mu$mol m$^{-2}$ s$^{-1}$) *versus* leaf temperature ($T_{leaf}$, °C). Each point represents one $A$-$c_i$ curve ($n$ = 86).**

[Figure]

seasons ■ *Cool* ● *Warm*

species
● *A.aneura*  ● *C.columellaris*  ● *E.salmonophloia*  ● *E.scoparia*
● *A.hemiteles*  ● *E.clelandii*  ● *E.salubris*  ● *E.transcontinentalis*

**Figure 6: Bivariate linear regressions of the $c_i$:$c_a$ ratio (at ambient $CO_2 \approx 400$ µmol mol$^{-1}$) *versus* temperature ($T_{leaf}$, ˚C) for individual trees considering all data (a) and within species (b). Only significant regressions ($p < 0.05$) are shown. Each point represents one $A$-$c_i$ curve ($n = 86$).**

[Figure]

**Figure 7: Bivariate linear regressions of natural log-transformed** $V_{cmax}$, $V_{cmax25}$, $J_{max}$, $J_{max25}$, $R_{dark}$ **and** $R_{dark25}$ ($\mu$mol m$^{-2}$ s$^{-1}$) *versus* **leaf temperature** ($T_{leaf}$, °C) **within species** ($p < 0.05$). **Only significant regressions** ($p < 0.05$) **are shown. Each point represents one** $A$-$c_i$ **curve** ($n = 86$).

[Figure]

**Figure 8: Bivariate linear regression of $N_{mass}$ (mg g$^{-1}$) *versus* $P_{mass}$ (mg g$^{-1}$) for all data (black line, slope = 0.33, $R^2$ = 0.17). Each point represents one leaf ($n$ = 86).**

[Figure]

**Figure 9: Changes in the average foliar N:P ratio for each species between the cool and the warm seasons. Standard errors shown.**

We thank the reviewers for comments and suggestions that have helped to improve our manuscript. Our response is organized by addressing each comment one by one.

**Anonymous Referee #1**

**AR1.1.** I find that while the results from the photosynthetic measurements are interesting and C1 well analysed and discussed, the theoretical analysis and the link between measurements and theory needs more discussion, in particular in relation to the many linear assumptions made, all of which are hidden in the appendix.

We have moved the equations to the main text and provided further discussion there. The text now includes the specific rationale for the linear assumptions that we made.

**AR1.2.** The conclusions relating to acclimation and coordination are based on the slopes of regression lines of photosynthetic variables to temperature but there is insufficient detail in the paper relating to the results of the statistical analysis. Figures 3-5 are presented without any goodness of fit metrics or p-values for the individual lines.

We believe this comment refers to Figures 6 and 7. But here, only significant individual lines ($p < 0.05$) are shown, as is described in the Figure captions and in the Results section (3.2).

**AR1.3**. In addition, the authors assume a linear relationship between the log10 values of each variable and temperature, an assumption which is detailed in appendix A but not sufficiently discussed in the main text.

Linear regressions are used because the theoretical equations relating log-transformed traits to temperature are linear. This should now be clear from the revised text, in which the equations are presented up-front. We have also made a specific statement about this matter in Section 2.5 in order to make the point absolutely clear.

**AR1.4.** The coordination hypothesis states that the Rubisco and electron transport limited rates are co-limiting under average conditions, which is generally taken to mean that there is a change in the Jmax25 to Vcmax25 ratio and implicitly a change in nitrogen allocation inside the leaf. The authors make a linear approximation to solve for this co-limitation (eq. A3). This approximation removes the parameter Jmax25 from the calculation and its slope with temperature is calculated assuming proportionality to the slope of Vcmax25 and a ratio of the biochemical temperature response. While these approximations can be justified, I believe that a further discussion is needed as the resulting equations are difficult to match with the coordination hypothesis as this is generally understood.

The central quantitative evidence is the acclimation in $V_{cmax}$, which is consistent with theoretical predictions. Nonethless, our results are also consistent with a simple additional hypothesis about the acclimation of $J_{max}$, namely the maintenance of a

constant ratio between $V_{cmax}$ and $J_{max}$ at a prescribed temperature (e.g. 25˚C). Our revision has clarified these matters.

However, while the reviewer might possibly be right to suggest that "it is generally taken" that the co-ordination hypothesis is about the **partitioning** of leaf N to $J_{max}$ versus $V_{cmax}$, our understanding is that it only states that **the two limiting rates of photosynthesis tend to be co-limiting under average conditions**. Limitation by $J_{max}$ is not normally reached under natural conditions; at low light photosynthesis is limited by $J$ (not $J_{max}$), and at high light it becomes limited by $V_{cmax}$. **So the first-order prediction of the co-ordination hypothesis is simply that $V_{cmax}$ should acclimate to the average conditions**, and that is our first-order result.

The co-ordination hypothesis thus makes a stronger prediction about leaf N than merely its partitioning to different functions. It actually implies that there is an **optimal value** of leaf N. This has been proposed in a number of papers that we cite, starting back in the 1990s (e.g. Dewar, 1996, predicts an optimal canopy nitrogen using the maximum NPP hypothesis where gross photosynthesis is subtracted by maintenance and growth respiration) and elaborated and evaluated more recently e.g. by Maire *et al.* (2012) and Dong *et al.* (2017).

**AR1.5.** I would also suggest including all the equations in the main body of the text since they are necessary to the central message of the paper

Done.

**AR1.6.** The authors report the slope of the log10 of each measured parameter with temperature and compare this to the theoretical equivalent slope (Table 1) to reach the conclusion that the coordination hypothesis is valid. The more usual approach would be to calculate the theoretically predicted values of the photosynthetic parameters and plot these together with the measured values. The authors' approach is scientifically valid but given the multiple approximations and log values I find it hard to follow.

We agree that plots are easier to follow than tables; however, we do present partial residual plots – please bear in mind that to maximize statistical power we have analysed the data together in a generalized linear model, using species as factors (as explained in the Methods – Statistical Analysis section, and the captions). We tried various ways to present the data and we found that the approach we have adopted here was the most accessible. We do not agree that a simple comparison of predicted and measured values would be preferable, because it would not convey anything about the mechanistic basis for data-model agreement. Our presentation of relationships to temperature carries the stronger message that these relationships exist in the data.

**AR1.6.** Also, the fitted slopes for all parameters are calculated as log10(parameter) vs. temperature, while the theoretical slopes are ln(parameter) vs. temperature. I would suggest that the authors check their calculations and verify that these slopes are equivalent.

This was a slip in the original version. All slopes (observational and theoretical) were calculated using natural logarithms. We have changed the text and Figures to make sure this is clear now.

**AR1.7.** Changes in Vcmax values alone do not verify the coordination hypothesis - these can be caused either by acclimation or by changes in total leaf nitrogen. According to Fig. 8 there are large differences in the leaf N for some species, which can be caused by a number of factors apart from temperature acclimation, especially leaf ageing. I would be interested to see how the ratio of Vcmax (and/or Jmax) to leaf N changes seasonally, which would give a better indication of photosynthetic coordination.

We do not agree that change in total leaf N could meaningfully be a "cause" of changing $V_{cmax}$. The co-ordination hypothesis implies (as pointed out by many authors, see remarks above) that there is an **optimal leaf N** for any given set of average environmental conditions. Thus, acclimation involves changes in $V_{cmax}$ and potentially changes in leaf N as well.

Nonetheless, in response to this comment, we have carried out additional analyses on how the ratio of $V_{cmax}$ (and $V_{cmax25}$) to $N_{area}$ varies with growth temperature. The results are generally very similar to the analysis of $V_{cmax}$ (and $V_{cmax25}$), but they are significant for more species (7/8: all except the gymnosperm *Callitris columelaris*, for both $V_{cmax}$ and $V_{cmax25}$). We have clarified the point, and referred to this additional analysis, in our revised text. See also the graphs below:

[Figure]

**AR1.8.** While acclimation of respiration is a well documented and important process it is unclear how this links to the coordination hypothesis. Here the authors hypothesised that dark respiration scales linearly with Vcmax and will therefore follow the coordination hypothesis as well, but this is not necessarily the case in either models or reality and a better justification of why the variation in dark respiration should be linked with photosynthetic co-limitation is needed.

The reviewer is correct to indicate that to predict the acclimation of $R_{dark}$ to temperature from the co-ordination hypothesis requires an additional hypothesis. We have tried out the simplest, i.e. that $R_{dark}$ remains proportional to $V_{cmax}$. This logic, which is also used in many models, is now explicitly spelled out.

**AR1.9.** The authors should decide whether we are talking about 'coordination' or 'co- ordination'.

The spelling 'co-ordination' has been replaced by 'coordination'.

**Anonymous Referee #2**

**AR2.1.** My main comment is that the discussion is very thin. It could use more substance and less reiterating the results. What do you make of the considerable spread in the data? Why do many species in Figure 6 not show the expected response, even if the pooled data does? There's a lot more here to discuss than is currently covered.

We have expanded the discussion explaining light limitation of $J_{max}$ as well as relationships of photosynthetic traits to nitrogen and temperature across the two seasons. We have also expanded the discussion of implications for modelling. However, we do not have any useful information on the causes of differences among species. This is normal for trait analyses! Our goal has been to uncover general, first-order relationships. At this stage it is not surprising that not every species conforms to a universal pattern.

**AR2.2.** There are a number of studies that have measured Vcmax and Jmax at multiple times across a season in the literature (Baldocchi has a few, for example). These should be acknowledged in the intro. Similarly, there should be a citation to Way and Yamori 2013 who found no change in Vcmax25 in a meta-analysis of plants grown at different temperatures.

We are have now included the suggested citations. A study from 2000 by Wilson and collaborators, including Baldocchi, was cited as example of data on the responses of photosynthetic traits on ecologically relevant time scales. We have also included a mention of Way and Yamori's differing findings in the Discussion.

**AR2.3.** Why was Rdark measured after only 5 mins in the dark? This is usually measured after at least 20 and often 30 minutes of darkness to get a true estimate of dark respiration.

This was a time-saving compromise to allow four or five replicate curves per machine per day, based on our experience that stable results are commonly obtained after 5 minutes. Moreover, this quick estimate should still be superior to the common practice of deriving $R_{dark}$ as one of the parameters in a curve-fitting routine. We have added a comment to this effect.

**AR2.4.** What VPD were the measurements made at? If the summer VPD is higher, gs will be reduced, which will lower the Ci/Ca ratio, but this isn't necessarily a temperature effect per se.

There is indeed a systematic difference between VPD in the two seasons. The average VPD value during the warm season was 1.13 kPa, and during the cool season 0.45 kPa. The average *leaf-to-air* VPD (i.e. corrected to leaf temperature) during the

warm season was 2.5 kPa, and during the cool season 1.44 kPa. However, there was very little difference in stomatal conductance at light saturation: (0.064 – 0.082 – 0.101) (lower quartile – median – upper quartile) mol m$^{-2}$ s$^{-1}$ for the warm season, and (0.057 – 0.078 – 0.085) mol m$^{-2}$ s$^{-1}$ for the cool season. We have added a note on this.

**AR2.5.** Figure 2 - why were the fits forced through the origin and how does this affect the slopes? Is it a minimal effect?

Both slopes are significant if not forced through origin. However, an intercept for this relationship is (a) extrapolated, and (b) makes no biological sense. We have added a note on this.

**AR2.6.** Lastly, while I appreciate the use of the log-transformed data to get linear slopes, I'd like to see the "real" data, at least in the SI. This makes it much easier to see the values measured and compare the data with the majority of other studies that report Vcmax and Jmax values against leaf temperature.

The real values are in fact presented in the Results section, and in Figure 2. Raw data are also available on the TERN portal as mentioned in the text.

Technical comments

**AR2.6.** Page 2, Line 13 - please clarify what "these" refers to - Vcmax and Jmax, yes?

Yes. We have clarified this now.

Page 9, Line 7 - the relationship between Ci/Ca and photosynthetic capacity could also be because higher photosynthetic capacity (at a constant gs) reduces Ci. Cause and effect can't be determined.

We have added a note on this alternative explanation in Discussion (section 4.4). It assumes that higher $V_{cmax}$ means more photosynthesis.... but if the coordination hypothesis is correct, then this assumption is incorrect.

If all the gas exchange is determined with a Licor IRGA, how are the parameters being reported in units of electrons and O2? Jmax and Rdark should be in units of CO2 per area per time.

We have amended the units as suggested.

**Anonymous Referee #3**

**AR3.1.** What is the role of phenology / specifically leaf age here, there is a need to discuss this either in the introduction and or discussion, i.e. there might be confounding phenological and thermal acclimation effects in the presented results. What is the life time of a leaf in this semi-arid evergreen woodland? -Related to the above, the manuscript provides an explanation of changes in N:P ratios from cold to

warm season in Fig 8 however it does not explain how these changes happened, how did leaf N and P changed and how this might be related to leaf age? It would be good to add a plot showing individual values of leaf N and Leaf P in the cold and warm season.

Three factors militate against any important role for phenology and leaf ageing in our results. First, the generally long (**5-10 years**) lifetime of leaves in this ecosystem, meaning that ageing proceeds slowly; second, there is **continuing turnover of leaves** through the year in this ecosystem; and third, we sampled **the youngest fully expanded leaves** in both seasons.

**AR3.2.** Equation 1 is used to estimate Vcmax and Jmax at 25C. During the warm period (unclear time of day A-Ci curves where taken) Vmax at T could be either in the optimum or beyond the optimum temperatures, thus it is possible that the peaked temperature response might be more appropriate. If this was the case, how is this likely to affect the results? Also, how does the choice of Ha (Medlyn et al 2002) value affects the results. According to Hikoska et al (2006) there is a relationship between activation energy of Vcmax and growth temperature. Similar comments apply to the use of equation A2 to determine the slope of Vcmax and temperature presented in Table 1 under the kinetic approach. Is the slope sensitive to the choice of Ha but most important are the slope values robust when estimated with the peaked temperature response for Vcmax and Jmax.

In theory $T_{opt}$ can affect the calculation of $V_{cmax}$ and $J_{max}$. However, Medlyn et al. (2002), Kattge & Knorr (2007), and others have found very good correlations between $T_{opt}$ and mean daily temperature. We measured $A$-$c_i$ curves between morning and early afternoon, avoiding the hottest part of the day. Therefore, it is unlikely that any of our measurements were carried out above the optimum temperature. We have added a comment to this effect.

Regarding the activation energy of $V_{cmax}$, there have been reports of a sensitivity to growth temperature but any such effect appears to be small and many studies have failed to find any such effect (e.g. Kattge & Knorr, 2007). No information was available on $H_a$ for the species sampled and so (in common with much of the ecophysiological literature) we adopted a generic *in vivo* value. We recognize that the activation energy of $J_{max}$ is much more variable with growth conditions.

**AR3.3.** Leaf dark respiration measurements were taken after only 5 minutes of leaves being in the dark. Protocol for Rd estimates is at least 30 min in the dark (Atkin et al 2000; Atkin et al 1998) as it takes about 15-20 minutes for post-illumination respiration to stabilize with time increasing with decreasing temperature. How does this affect your measurements of Rdark and acclimation results?

This was a time-saving compromise (see our response under AR2.3 above).

**AR3.4.** On the implications for modelling section it would be very relevant to apply the Kattge & Knorr (2007) formulations and compare to your data set and predictions by the optimization approach used in this study.

See our response to AR3.2 above.

**AR3.5.** Is the data from this study consistent with the Vcmax25 prediction derived by Scafaro et al (2017)

Yes. This was mentioned in the original text, but the reference was missing in the bibliography. We have rectified this omission.

**AR3.6.** Either in the introduction or in the methodology, it would be good to include a graphic explaining the change in temperature responses to illustrate what acclimation is, i.e. temperature response shifts forward and therefore values at 25 C decline, you could illustrate also where in the curve are the leaf temperature values are during the cold and warm season.

We have provided the suggested graphic, as the new Fig. 1.

**AR3.7.** It would be useful to include a figure of the mean diurnal cycle of air temperature during the warm and cold seasons but also provide an idea of when the A-CI curves were taken and under which RH, VPD conditions. If RH & VPD conditions differ, what are the implications.

The Methods section already provides information on the diurnal cycle of temperature in both seasons, and the timing of our measurements. Regarding the potential effects of VPD, please see our response under AR2.4 above.

**AR3.8.** P10, L 25 Can you clarify in the text why the acclimated slope of Jmax to leaf temperature was estimated as the acclimated slope of Vcmax minus the difference of the kinetic slopes of Vcmax and Jmax (this might also be affected by peaked temperature response)

We have now explained exactly what we are doing here in the revised text. It is a simplification, as described in our response to AR1.4 above, but we hope now it is clear.

**AR3.9.** P3 L2 –can elaborate here and explain homeostasis

We have changed the wording here, to be more explicit, and avoided the term homoeostasis.

**AR3.10.** P6 L22 Is this Tleaf measured by the Licor or an independent measurement? If yes would be good to mention it in the methods section

$T_{leaf}$ is the leaf temperature as measured by the LiCor. We have noted this in the text now.

**AR3.11.** P7 L26-29 These values were not really shown as it was all logged transformed, would be nice to show the data.

The values are presented in the Results section and in Figure 2. Raw data are also available on the TERN portal, as mentioned in the text. Since the manuscript already includes nine Figures and one Table, we prefer to keep the reader's attention focused on the main results.

**AR3.12.** The sentences comparing values to dessert plants and mesic perennial species could be more specific and include typical values for those vegetation types otherwise is all very generic and less informative.

We have included some example values.

**AR3.13.** P8 L6 but 'lower allocation of N to Rubisco' has not been demonstrated here.

We agree, but this is presented in the text here as a prediction, not as a fact!

**AR3.14.** P8 L9, need to mention the role of leaf age /phenology, maybe here good to show N values change and use this to support some of the sentences on this paragraph.

Please see our response to AR 3.1.

**Short Comment #1**

**SC1.1.** Based on their data or via model simulations, suggest how the ecosystem models can be improved. That is, if you were to use an ecosystem model, how would the parameters that you measured change with time in the model. In my view, coordination hypothesis has already been implemented in some ecosystem models.

We have added some wording to address this point in the Discussion.

**SC1.2.** You have the seasonal data and you just connect two points in Fig. 8. First in my view, this does not seem right. It would be nice to show better the temporal variation of the parameters for these evergreen species. My main concern here is to specify how much is the variation in the parameters of these evergreen species due to the different seasons e.g. 10%, 20%, etc.

We were not monitoring the species through a whole year, and so it is not possible to provide what is asked for here. However, we have provided some indication of the relative magnitude of seasonal changes in parameters.

**Associate Editor**

**AE.1.1.** Answer 1.4: I find this answer only partially convincing because if I follow your line of argumentation, in light saturation, photosynthesis is not limited by Vcmax, but by Vc, as there is still a Ci dependence to be respected.

Of course the $c_i$ dependence of photosynthesis is important, but it doesn't alter our argument. The coordination hypothesis indicates that under typical daytime conditions, the Rubisco-limited and electron transport-limited rates of photosynthesis should be approximately equal, **given a set of conditions that include $c_i$ (and its influence on both rates).** And because light is 'external' to the plant, the first-order prediction of the coordination hypothesis is that $V_{cmax}$ must adjust to the available light. We hope we have now made this clear in the text, while avoiding too much distracting complexity. See our response to AR 1.4 above.

**AE.1.2.** Answer to comment 2.1/2.2: This answer is too superficial for the open discussion. An outline of your responses to these comments would have been appropriate. Please be sure to provide more detail in your response letter

Please see our extended responses to AR 2.1 and 2.2 above.

**Recommendations to the editor: Report 2**

**R2.A:** I would like to thank the authors for their detailed response to my comments. The only issue that I do not think has been addressed sufficiently is the definition and implications of the coordination hypothesis which lies at the basis of this paper. While the authors correctly say that the coordination hypothesis simply states that the two limiting rates of photosynthesis, Av and Aj should be equal, there are two opposing interpretations of this statement. The first one, which the authors also subscribe to, is that the Jmax/Vcmax ratio remains constant and the leaf N changes (Chen et al. 1993; Maire et al. 2012). The alternative is of course that the leaf N remains constant and the Jmax/Vcmax ratio varies (Ali et al. 2015; Medlyn 1996; Quebbeman and Ramirez 2016). These two contrasting approaches should be mentioned in the introduction and the methods and which of these the authors use in their model analysis.

We have made the inclusion suggested.

As a side note, the (Dewar 1996) study which the authors mention in their reply, does indeed predict an optimal canopy nitrogen but does so using the maximum NPP hypothesis, which is distinct and must not be confused with the coordination hypothesis.

We included a note as suggested on our response to AR.4